**communications** engineering

# A semi-transparent thermoelectric glazing nanogenerator with aluminium doped zinc oxide and copper iodide thin films
Mustafa Majid Rashak Al-Fartoos, Anurag Roy ✉, Tapas K. Mallick & Asif Ali Tahir ✉

To address the pressing need for reducing building energy consumption and combating climate change, thermoelectric glazing (TEGZ) presents a promising solution. This technology harnesses waste heat from buildings and converts it into electricity, while maintaining comfortable indoor temperatures. Here, we developed a TEGZ using cost-effective materials, specifically aluminium-doped zinc oxide (AZO) and copper iodide (CuI). Both AZO and CuI exhibit a high figure of merit (ZT), a key indicator of thermoelectric efficiency, with values of 1.37 and 0.72, respectively, at 340 K, demonstrating their strong potential for efficient heat-to-electricity conversion. Additionally, we fabricated an AZO-CuI based TEGZ prototype (5 × 5 cm²), incorporating eight nanogenerators, each producing 32 nW at 340 K. Early testing of the prototype showed a notable temperature differential of 22.5 °C between the outer and inner surfaces of the window glazing. These results suggest TEGZ could advance building energy efficiency, offering a futuristic approach to sustainable build environment.

In modern architectural designs, buildings account for ~40% of global energy consumption, with windows responsible for 20–40% of the energy wasted within these structures. This issue is expected to worsen as global development continues, leading to more buildings being constructed and an increasing number of structures featuring extensive glass facades[1]. Integrating sustainable materials into buildings can decrease energy losses and simultaneously generate energy. There are many attempts to implement solar photovoltaics in windows to produce electricity[2,3]. However, they are expensive, require sunlight, and their performance could be affected by dust, heat, etc.

Reducing the cooling and heating demands of buildings is crucial to pivot towards greener initiatives[4]. This reduction can be achieved by bolstering energy efficiency, which involves reducing incoming solar radiation and heat transfer from the interior to the exterior[5]. Introducing sustainable materials like low-emissivity coated glass and aerogel glazing offers a solution[6]. Moreover, the development of architectural windows capable of intelligently controlling indoor solar radiation through changes in their optical transmittance holds great promise for reducing energy consumption in buildings[7]. Recently, energy-efficient smart window technology has garnered noteworthy scientific attention, leading to the exploration of innovative materials and their integration with practical techniques to achieve desired multifunctional properties[8,9]. Such as energy-saving glazing systems predominantly employ multilayer, vacuum, polymer-dispersed, thermochromic/electrochromic, low-emission, or phase change materials[10].

However, while these materials enhance insulation, they neither generate nor conserve energy. Enter thermoelectric (TE) materials—renewable energy sources capable of directly converting heat to electricity. Various economic factors, such as the long-term uncertainty in the price of fossil fuels, are also stifling the development of TE devices on a commercial scale. Yet, integrating a thermoelectric generator (TEG) in glazing systems offers a dual advantage: minimising energy wastage and simultaneously harnessing this waste energy to produce electricity. TE materials stand out as a sustainable energy conversion approach. They can directly transform heat into electricity, leveraging the Seebeck effect. Moreover, TE devices offer the benefits of compactness, noise-free operation, ease of use, and low maintenance due to their lack of moving parts[11]. Implementing TE materials in windows offers the opportunity to generate electricity through harvesting the inherent temperature gradient between the interior and exterior of a building[12].

Solar Energy Research Group, Environment and Sustainability Institute, University of Exeter, Penryn Campus, Penryn, Cornwall, TR10 9FE, UK.
✉e-mail: a.roy30@exeter.ac.uk; a.tahir@exeter.ac.uk

Thermoelectric glazing (TEGZ) is critical for increasing building energy efficiency. They turn heat into electricity by leveraging temperature variations and lowering total energy consumption. TEGZ in semi-transparent form, enables windows to provide natural light, thermal insulation, and clean energy production all at the same time, contributing to sustainable and environmentally friendly construction solutions. The efficiency of TE materials in converting energy depends on temperature differentials and a dimensionless figure of merit (ZT). The ZT value shows an inverse relationship with thermal conductivity ($k$) and a direct relationship with the square of the Seebeck coefficient ($S$) and electrical conductivity ($\sigma$), together constituting the power factor (PF). However, many high-performance TE materials are deemed toxic, expensive, and rare.

For TEGZ to be viable, it must retain transparency to ensure both visual and thermal comfort within buildings. Recent advancements demonstrated a TEG module with impressive transmittance, achieving over 81% using p-type poly(3,4-ethylenedioxythiophene): poly(styrene sulfonate) and n-type indium tin oxide (ITO)[13]. Moreover, the operational temperature range of TEGZ needs to be compatible with window ambient conditions. For instance, a p-type TE material has been obtained using Te-embedded $Bi_2Te_3$, which displayed remarkable ZT at 370 K, attributed primarily to reduced $k$ via heightened phonon scattering[14]. The scalability of the production process is also essential to accommodate vast window areas. For example, chemical batch processing could converted $Bi_2Te_3$ nanowires into ink suitable for glass substrate printing[15].

A few ground-breaking attempts have already surfaced in the realm of TEGZ. For example, a 132.25 $cm^2$ TEGZ prototype created by using Plexiglas panels embedded with nanopowders derived from mechanical alloying[16]. Conversely, a developed 100 $cm^2$ TEGZ covered with a wavelength selective film incorporated with $Bi_2Te_3$-based TEGs[17]. While these are impressive strides, challenges such as material cost, limited transparency, and heat transfer remain. Previous studies have achieved substantial TEGZ performance using ZnO and CuI[18,19]. However, production complexities and low efficiency continue to be limiting factors in this research.

ZnO and CuI are environmentally friendly and economically viable wide-bandgap semiconductors, making them suitable candidates for n-type and p-type TEG legs, respectively. They could be alternatives to the traditional TE materials such as lead or telluride. They are abundant in the Earth's crust, thus alleviating concerns regarding resource depletion and mitigating the environmental impact associated with mining and extraction. Tellurium, though rare at 0.001 ppm, is priced high at £1200/kg due to its TE efficacy. Conversely, copper, abundant at 70 ppm, is cost-effective at £63.39 for 1 kg[20]. This data aids in informed material selection and cost projections for TE device development. For example, a flexible p-type transparent TE materials based on CuI thin films, demonstrating remarkable room temperature performance with a ZT = 0.21 at 300 K[21].

The TE performance of ZnO, characterized by its ZT, is notably low near ambient conditions, registering a value close to $1.4 \times 10^{-4}$ at 350 K[22]. By 1000 K, this value only marginally improves to ~0.02. Moreover, ZnO has high electrical resistance. To address these limitations, researchers have predominantly focused on two central strategies: (i) interstitial doping with n-type elements such as Al, Ga, and In. Such a modification to ZnO aims to elevate its ($\sigma$). This, in turn, can boost PF. (ii) Nano-structuring techniques offer another avenue for performance improvement. For example, various morphologies of ZnO, such as nanoribbons, nanorods, nanoparticles, and nano shuttles, exist, with ZnO nanorods demonstrating optimal TE performance[23].

On the other hand, γ-CuI emerges as a promising contender for serving as a semi-transparent p-type leg within the framework of TEGZ owing to its remarkable combination of attributes. Characterized by high Seebeck coefficient and $\sigma$, copper, in conjunction with its low $k$ counterpart iodine, presents an enticing proposition. This distinctive feature can be primarily attributed to the presence of the heavy element iodine. Moreover, numerous methodologies are available to augment the performance of CuI. These techniques encompass doping, nano-structuring, carrier concentration

optimization, and strategies to diminish $k$ through the introduction of defects or by employing heavy element doping.

This study reveals the development of aluminium-doped ZnO (AZO) nanorods, synthesised through electrochemical methods, functioning as an n-type material. In addition, the γ-CuI derived via the successive ionic layer adsorption and reaction (SILAR) method serves as a p-type material. Together, they form a semi-transparent TEGZ designed for energy-efficient building environments. Our innovative prototype semi-transparent TEGZ fabrication techniques, involving the deposition of nanostructured AZO and CuI on fluorine-doped tin oxide (FTO) glass, represent cost-effective scalable solutions. Moreover, our adoption of FTO glass aligns with contemporary low-emissive glazing technologies such as Pilkington K-glass[24]. These techniques effectively yield a highly efficient TEGZ by modulating fabrication thickness and optoelectronic properties on enhancing their transparency or devising TEGZ designs that do not compromise transparency or indoor comfort. The comprehensive evaluations of TEG power output and the in-depth exploration of temperature distribution within the TEGZ provide a holistic understanding of its capabilities and potential.

## Methods
### Materials
$Zn(NO_3)_2.6H_2O$ from Fisher Scientific, $NaNO_3$ from across organic, HMTA (hexamethylenetetramine $C_6H_{12}N_4$) from merk, $Al(NO_3)_3 \cdot 9H_2O$, $Na[Cu(S_2O_3)]$, $CuSO_4$, $Na_2S_2O_3$, NaI from Thermo Scientific, FTO glass (NSG TEC 5) from, Pilkington. All the chemical reagents have been used with no further purification. All the solutions were prepared by double distilled water.

### Synthesis of pure and Al-doped ZnO nanorod thin film
The samples of ZnO NRs with different percentage of Al doping has been deposited on FTO glass by a standard three electrochemical deposition system[25,26]. The system contains from reference electrode (potassium chloride saturated Ag/AgCl electrode), a working electrode (FTO glass) and a counter electrode (Pt wire) as shown in Supplementary Fig. 1a. These samples were deposits from an aqueous solution containing 1.0 mM Zn $(NO_3)_2.6H_2O$, 0.1 M $NaNO_3$, 1.0 mM HMTA (hexamethylenetetramine $C_6H_{12}N_4$), and 0, 0.02, 0.03 or 0.06 mM of Al $(NO_3)_3 \cdot 9H_2O$. Before the deposit, all FTO substrates underwent thorough cleaning using ultrasonication with distilled water, ethanol, and acetone, then drying. The samples were deposited on FTO using a potentiation electrolysis procedure in two steps. The first step was electrolysing at −1.3 V vs. Ag/AgCl for 0.5 s to create a seed layer. Following that, ZnO NRs were synthesised by applying 1.0 V vs. Ag/AgCl for 30–120 min. The bath temperature of the electrochemical depositing was controlled at 80 °C by a thermometer throughout the electrodeposition. Following electrodeposition, the produced samples were washed with water, dried at room temperature, and then annealed in a tube furnace at a temperature range from (room temperature (21 °C) to 500 °C for 1 h. Electrochemical depositions were carried out with the help of an Autolab PGSTAT30 potentiostatic/galvanostatic linked to a running NOVA 2.1 software.

### Synthesis of CuI thin film
A copper iodide thin film was deposited by the SILAR method on FTO glass. The synthesis was carried out similarly as described in refs. 27–30. Before the deposit, all FTO substrates underwent thorough cleaning using ultrasonication with distilled water, ethanol, and acetone, then drying. The SILAR cycle procedure illustrated in Supplementary Fig. 1b contains four beakers. Beaker A contains an aqueous copper (I) thiosulfate complex $Na[Cu(S_2O_3)]$ solution, which acts as a cationic precursor. This solution is formed by mixing 0.25 M of $CuSO_4$ and 0.167 M of $Na_2S_2O_3$. Beakers B and D contain double-distilled water. Beaker C contains 0.05 to 0.1 M of aqueous NaI solution, which acts as an anionic precursor. When the substrate was dipped for 5–20 s into beaker A, a monolayer of copper ions $Cu^+$ started to be absorbed on its surface. Then the substrate is dipped into beaker B for 10 s to remove the other ions. After that, the substrate was dipped into beaker C

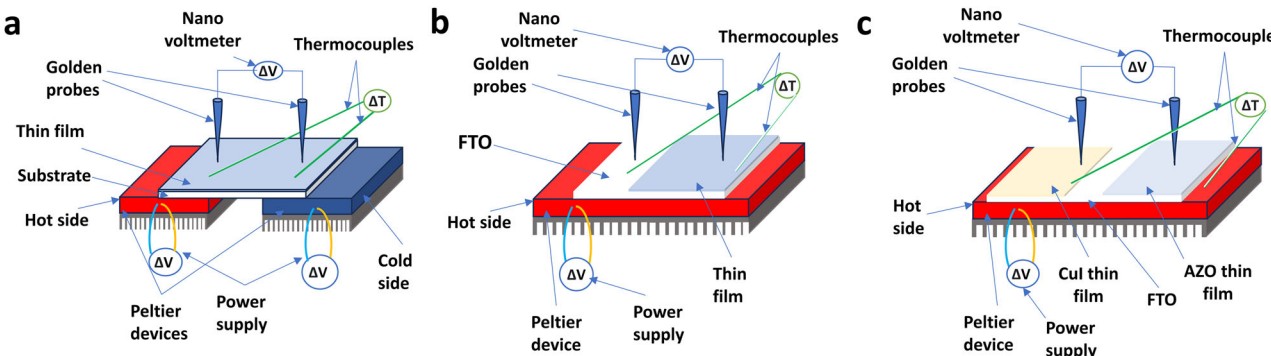

**Fig. 1 | A schematic representation of the custom-designed device employed for specific thermoelectric (TE) thin film and thermoelectric generator (TEG) measurements.** Three sections of the illustration delineate the apparatus' capability: **a** The section labelled "in-plane Seebeck" shows the design elements essential for measuring the in-plane Seebeck coefficient. **b** Moving on, the "cross-plane" section illustrates the components designed to measure the cross-sectional Seebeck by measuring the temperature gradient and the associated voltage difference. **c** Lastly, the "power output" segment delineates the setup employed to quantify the electrical power generated by the TEG under test conditions.

for 20 s to react the copper ions with iodine. Finally, the substrate is dipped into beaker D for 10 s to remove the loosely bound particles and ions. This cycle was repeated 25–40 times. After cycles were completed, the substrate was dried in air and then annealed at temperatures ranging from room temperature (21 °C) to 300 °C. The SILAR depositions were carried out with the help of the Ossila Dip Coater.

## Materials characterisation

The characterization of the crystal structure for the thin film was done by a Siemens D5000 automated powder diffractometer (Co-Kα irradiation, 40 kV/40 mA). XT Nova Nanolab 600 was used to perform scanning electron microscopy (SEM) to study thin film coating microstructure. Moreover, Tescan Vega3 was used to perform energy-dispersive X-ray analysis spectroscopy (EDS) on the thin film coating, which allowed for the mapping of its microstructure and elemental composition. The Bruker Innova atomic force microscope (AFM) was used to study surface profiles and measure thin film thickness. The tapping mode and ambient conditions were used to obtain the AFM image. In addition, morphology and structural analysis included high-resolution transmission electron microscopy (HR-TEM) and selected area electron diffraction (SAED) using the JEOL JEM-2100F TEM operating at 200 kV. A Perkin Elmer LAMBDA 1050 UV/Vis/NIR spectrophotometer was used to study the transmittance spectra of the AZO and CuI films deposited on FTO substrate samples from 200 to 800 nm. The optical energy band gap ($E_g$) was calculated by absorbance spectra, as in previous studies[31,32] by Tauc's relation (Eq. 1) as below:

$$(\alpha h\nu)^2 = c(h\nu - E_g) \quad (1)$$

Where $\alpha$ is the absorption coefficient, $h\nu$ is photon energy, $E_g$ is the band gap energy and $c$ is constant which can be determined by using the following Eq. (2):

$$(2.303*A*1240/\lambda)^2 = c(1240/\lambda - E_g) \quad (2)$$

Where $\lambda$ is wavelength, A is absorbance taken from the absorption spectra of the thin film.

$\sigma$ was determined by employing an Osilla four-point probe, while a Peltier device with voltage control was utilized to modulate the film's temperature. To determine $S$, induced TE voltages in response to temperature gradients ($\Delta T$) along the deposited TE films on FTO were measured, as previously described in refs. 18,19,33. The $k$ measurements were performed using a C-Therm Trident system with a modified transient plane source. The method applied was the Modified Transient Plane Source technique following ASTM D7984-16. For each sample, the mean value of five measurements of $k$, taken at 1-minute intervals, was used to assess the

stability of the readings. Besides, each sample was measured five times for all other measurements.

## Device fabrication and prototype set up for thermoelectric performance measurement

To determine the in-plane $S$ of the AZO and CuI thin films, a temperature difference was created by placing the samples on hot and cold sources. The power measurements were employed to assess the voltage differences (Fig. 1a). This setup involved using two Peltier units connected to independent voltage controllers to apply the temperature gradients across the film samples. Temperature gradients were monitored using chrome-alum (K-type) thermocouples. The nano-power measurement device recorded voltage differences by positioning two gold electrodes, one on the hot side and one on the cold side of the samples. The $S$ was calculated by dividing the measured voltage change ($\Delta V$) by the applied temperature gradient ($\Delta T$). This measurement was performed in the near-room temperature range of 275–340 K. Subsequently, PF for the CuI and AZO thin films was calculated as Eq. (3) below:

$$PF = S^2\sigma \quad (3)$$

The device design (Fig. 1b) determines the cross-plane $S$. In this setup, the thin films are selectively deposited on FTO substrate, covering ~75% of its surface area, leaving the remaining portion available as an electrode. Two golden probes are strategically positioned for measurements: one in contact with the upper surface of the thin film and the other in contact with the FTO substrate. The probe in contact with the upper surface measures the voltage on the cold side, while the probe in contact with the FTO substrate measures the voltage on the hot side. The difference in voltage between these two probes represents the thin film output voltage. Two (K-type) thermocouples were used to measure heat differences between the upper surface of the thin film and the FTO. Prior each measurement a calibration was inducted by using pure copper metal (99.6%) as p-type references and pure nickel metal (99.6%) as n-type. Supplementary Table 1 represents a comparison between our homemade device result and with previous reported one[34].

To measure the power output of the TEG, which consists of p-type (CuI) and n-type (AZO) deposits on FTO, was placed on the Peltier device. The nanovoltmeter was used to measure the output voltage and current by the two probes placed on the AZO and CuI thin films. Two thermocouples are used to measure the temperature differences across the TEG by placing one on the upper surface of the TEG, and another one is on the heat source (Fig. 1c). The output power ($P_{out}$) is measured by the Eq. (4):

$$P_{out} = VI \quad (4)$$

The calculated power output performances, current–voltage ($I$–$V$) and power–voltage ($P$–$V$) characteristic curves were recorded during the varying temperature gradient in the n-p module TEG. The open circuit voltage ($V_{OC}$) is calculated according to Eq. (5) described by Yang et al.[21].

$$V_{oc} = S\Delta T \tag{5}$$

The voltage output ($V_{out}$) which is the voltage at the load resistance terminals to acquire the maximum power ($P_{out}$), was calculated using Eq. (6):

$$V_{out} = n(S_p - S_n)\Delta T - IR_{in} \tag{6}$$

Where $n$ is the number of TE elements, $S_p$ is the Seebeck of p-type, $S_n$ is the Seebeck of n-type, and $R_{in}$ is the internal resistance of the TE element.

Finally, the output power is calculated by the Eq. (7):

$$P_{out} = n(S_p - S_n)\Delta TI - I^2 R_{in} \tag{7}$$

## Results

### Optical and microstructural characterizations of AZO-CuI thin films

The XRD patterns of ZnO and AZO thin films with different Al-doping levels deposited on FTO glass substrate were achieved to study the thin film crystal phase (Supplementary Fig. 2a). The diffraction peaks shown by the square in the XRD patterns are the $SnO_2$ diffraction peaks of the FTO glass substrate. Other diffraction peaks show the crystalline nature of the ZnO, which matches the pattern with the standard (JCPDS card no. 00-036-1451).

The XRD pattern reveals the peaks of the (100), (002), (101), (102), (110), (103), and (112) planes of ZnO's hexagonal wurtzite-type structure[35]. The well-defined peaks in the XRD pattern indicate no peaks corresponding to the other compounds, such as the Al peak or $Al_2O_3$ phase. This indicates that Aluminium atoms have replaced Zinc atoms in the ZnO lattice. Moreover, Al ions might also be segregated to the non-crystalline areas along the grain borders, forming Al–O bonds or occupying the gaps between the ZnO lattice points.

The properties of ZnO thin films were considerably improved through doping. The XRD pattern reveals that the (002) and (101) diffraction peaks exhibit the highest intensities. After doping with aluminium, this peak shifts towards a lower angle, which can be attributed to the substitution of $Zn^{2+}$ ions by $Al^{3+}$ ions at ZnO lattice sites. The intensity of the peaks rose as the level of Al-doping increased up to 3%. However, raising the doping to 6% reduced the intensity of the peaks dramatically. This shows that larger doping levels negatively affect the crystal structure of ZnO. Al-doping leads to the enhancement of ZnO's well-ordered crystalline structure up to a specific doping limit.

The lattice constant c increased from 5.16 to 5.24 Å with Al doping up to 6%. The crystallite size was calculated using the Scherrer formula. It shows that doping decreases crystallite size from 36.66 to 22.36 nm with Al doping up to 6%, as illustrated in Supplementary Table 2.

In addition, it's noticed that $S$ increases with doping, where the crystal structure enhances until there is a certain level of doping and it reduces with the collapse of the crystal structure. This enhancement of the $S$ is attributable to the refinement of crystal structure[36].

There are three forms of CuI: α-CuI, β-CuI, and γ-CuI[37]. The most stable of these phases at ambient temperature is the γ phase. The crystalline nature of the γ-CuI compound was confirmed by X-ray powder diffraction (XRD), which showed that the synthesised compound had a pattern very similar to the pattern published in the literature (JCPDS card no. 82-2111) for γ-CuI[38]. The well-defined peaks in the XRD pattern indicate a crystalline structure, with slight broadening due to the nanosized nature of the synthesised compound. Regardless of parameter modifications, all diffraction peaks were ascribed to γ-CuI, and no impurity has been observed.

Analysis of the XRD data revealed a face-centred cubic (FCC) structure with an average lattice parameter (a = 6.067 Å). According to the Scherrer equation, the CuI crystallite diameter increased with rising NaI concentration, ranging from 23 to 28.6 nm as shown in Supplementary Fig. 2b. In addition, it expanded with elevated annealing temperatures, increasing from 25.7 to 38.4 nm as shown in Supplementary Fig. 2c. The crystallite size and lattice parameter of CuI thin films, obtained from XRD patterns, were compiled in Supplementary Table 3. This data encompasses variations in NaI ratios and annealing temperatures, providing insights into the structural characteristics of the films.

Furthermore, the XRD analysis revealed that the CuI films exhibited an intense peak at 2θ 25.24°, corresponding to the (111) plane orientation. The peak intensities of the planes increased with the addition of iodine, suggesting an improvement in the crystallinity due to iodine incorporation. However, after a certain amount of iodine, the peak intensities decreased. Moreover, the peak intensity increases with annealing temperature until 150 °C, when it decreases. Interestingly, samples with high $S$ exhibited high peak intensities, indicating that it could be an increase in crystal structure and an enhancement of periodicity in the CuI layers or a high concentration of defects that are not affecting the overall crystal structure, potentially enhancing the films' physical properties[37,39,40].

Figure 2a–c shows scanning electron microscope (SEM) images exhibiting the AZO nanorod arrays with varying Al dopant levels. The images reveal a thin film characterised by a high-density homogenous morphology. The nanorods are well-aligned, displaying hexagonal configurations with random orientations at their tips, corroborating the XRD results, which attest to the polycrystalline nature. The nanorods possess an average diameter ranging from 150 to 180 nm, with their surfaces presenting a smooth and continuous appearance. Interestingly, a subset of these nanorods possess diameters smaller than 40 nm. Notably, at the 2% doping level, the surface exhibits a similarity to that of pure ZnO characterised by a granular, highly dense, and compact structure (Fig. 2a). As the Al doping concentration increases, a marked morphological transition is observed. The diameters of the nanorods discernibly decreased, paralleled by a shift in their shape to more tapered forms and higher randomness in orientation, as evidenced in Fig. 2b, c. Nanorods, as a low-dimensional TE material, have gained substantial attention in recent researches. Their enhanced energy conversion performance can be attributed to the energy filtering effect and quantum confinement. These properties make them particularly promising in the realm of TE materials.

Figure 2d–i shows SEM images that delineate the morphological features of the synthesised (CuI) thin films. The thin film exhibits a uniform composition, although its surface is notably rough and incorporates particles that form cube-like structures. In Fig. 2d, when the iodine concentration is 0.05, the particle size appears small, yet the surface remains particularly rough and uneven, displaying an inhomogeneous texture. This roughness gradually diminishes as the iodine concentration is increased. Notably, an increase in NaI concentration from 0.075 (Fig. 2e, f) to 0.1 (Fig. 2g) leads to larger particle sizes from 40 nm to ~200 nm, respectively. Furthermore, when the annealing temperature increases, the average particle sizes increase from ~40 nm at 150 °C (Fig. 3h) to 300 nm at 200 °C (Fig. 2i). This suggests that higher annealing temperatures contribute to the enlargement of particle sizes within the film.

Upon further analysis of Fig. 2j, elucidation of the bright field image obtained from TEM revealed a rod-shaped morphology typified by an approximate diameter of ~100 nm and an average length of ~1 μm. This distinct morphology was consistently observed in the 3% AZO sample. HR-TEM image corresponding to 3% AZO confirm the highly crystalline nature of individual zinc oxide rods, with interplanar spacings corresponding to the (002) reflection, as depicted in Fig. 2k. Notably, upon exceeding an aluminium doping concentration of >3%, the morphology of the 3% AZO sample transitioned to include rod-like whiskers and irregular-shaped particles. The SAED pattern associated with this morphological transition reveals [002] zone spots, indicative of excellent high crystallinity as depicted in Fig. 2l. In Fig. 2m, the TEM bright filed image analysis

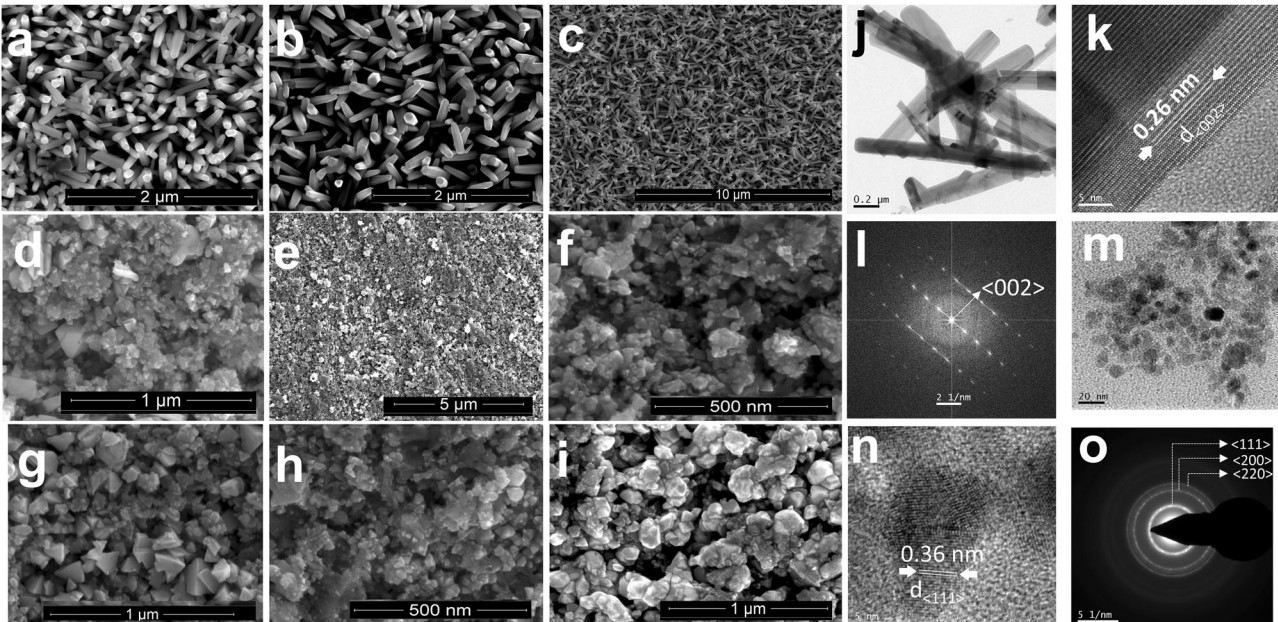

**Fig. 2 | Optical micrographs illustrating variations in aluminium-doped zinc oxide (AZO) and CuI thin films based on dopant and annealing conditions.** Scanning electron microscopic images where (**a**–**c**) represent AZO nanorod arrays with Al dopant concentrations. **a** 2%, **b** 3%, and **c** 6%, respectively. **d**, **e** Shows CuI thin films with NaI concentrations. **d** 0.05 and **e** 0.075. **f** The image provides a low-magnification view of the 0.075 NaI CuI thin film. **g** Features the film with 0.1 NaI concentration. **h**, **i** Temperature annealing effects on the CuI thin films. **h** 150 °C and **i** 200 °C. **j** Transmission electron microscopy (TEM) brightfield image of 3% AZO. Corresponding high-resolution TEM (HRTEM) and selected area of diffraction (SAED) images are presented in (**k**) and (**l**). **m** TEM brightfield image of CuI. Corresponding HRTEM and SAED images are presented in (**n**) and (**o**).

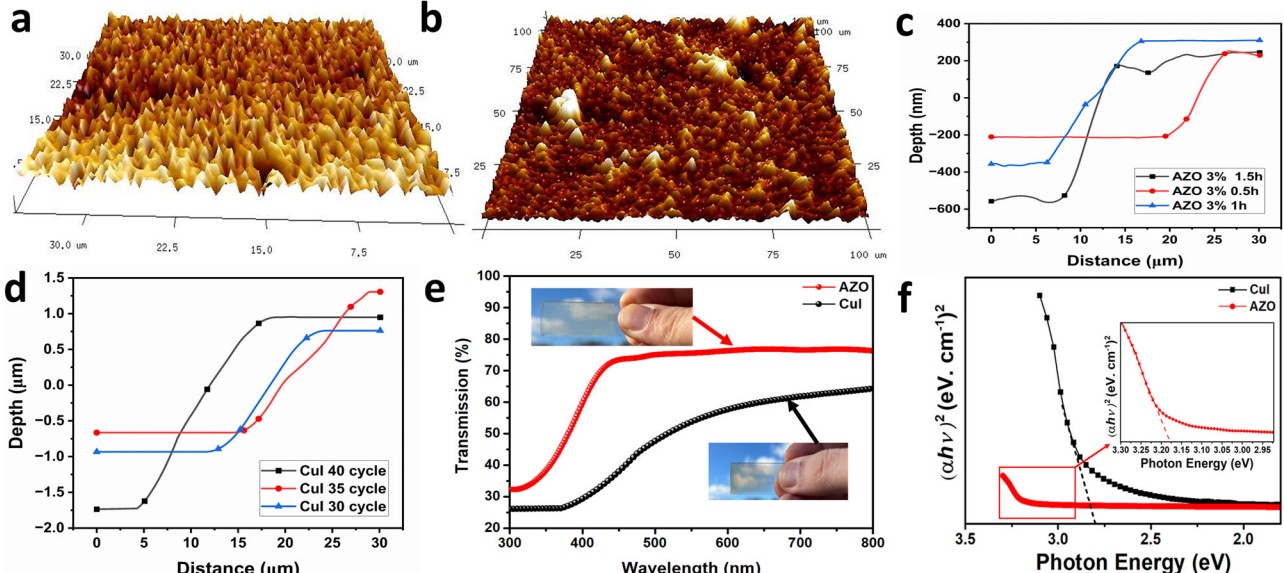

**Fig. 3 | AFM images, depth profile and assessment of optical characteristics for various ZnO-CuI thin films. a** 3D AFM image shows the topography of an aluminium-doped zinc oxide (AZO) thin film with a doping concentration of 3% **b** 3D AFM image shows detailed depiction CuI with a NaI concentration of 0.075. **c** The depth profile of AZO samples that have been subjected to various electro-chemical depositing durations. **d** The depth profile of CuI samples subjected to different numbers of dipping cycles. Collectively, these illustrations offer insightful details on the morphology and topographical nuances of the respective materials under varied conditions. **e** The transmission spectrum of the optimize AZO and CuI thin films that has the highest power factor. A photograph of the optimized AZO and CuI thin film against sky. **f** The optical bandgap energy of the highest power factor observed among AZO and CuI thin film samples.

unveiled the spherical morphology of the synthesised CuI nanoparticles. These TEM images illustrate that the CuI NPs exhibit a uniform spherical shape with an average size falling within $30 \pm 2$ nm. Further scrutiny through HR-TEM imaging revealed distinct lattice fringes characterized by a d-spacing of ~0.36 nm, corresponding to the (111) crystal plane of synthesised CuI NPs, as shown in Fig. 2n. In addition, the SAED pattern corroborated the high crystallinity of the CuI NPs, showcasing sharp diffraction spots indicative of their well-defined crystalline nature, as shown in Fig. 2o. Notably, the SAED pattern exhibited prominent peaks corresponding to the crystallographic planes of CuI, thereby affirming the purity and crystalline structure of the synthesised nanoparticles. These findings were further supported by XRD analysis, as presented in Supplementary

Fig. 2b, c, which demonstrated a close match between the experimental peaks and the standard γ phase of CuI.

The morphological characteristics of AZO and CuI thin films deposited on FTO substrates were systematically evaluated using AFM. Figure 3a exhibits the 3-dimensional (3D) AFM micrographs of the AZO thin films, demonstrating a prominent columnar structure oriented perpendicular to the substrate. This observation substantiates the c-axis orientation and agrees with the XRD data.

Conversely, Fig. 3b illustrates the three-dimensional AFM representations of the CuI thin films. These micrographs reveal a columnar-like morphology-oriented perpendicular to the substrate, albeit with a slightly randomized orientation. It is noteworthy that variations in luminescence intensity correspond to changes in height on the order of the film thickness, as determined through AFM depth profiling.

Further elucidation is provided by Fig. 3c, d which delineate the depth profiles for CuI and AZO, respectively, enabling the measurement of thin film sample thicknesses. The calculated thicknesses for the respective samples, as derived from these depth profiles, are comprehensively tabulated in Supplementary Table 4. The film thicknesses of AZO increased from 470 to 800 nm with an increase in deposition time from 0.5 to 1.5 h, respectively. Meanwhile, the thickness of CuI thin films increased with an increase in dipping cycles, from 1.7 μm at 30 cycles to 2.68 μm at 40 cycles.

Further, to confirm the Al existences as a dopant in ZnO, the EDS of AZO and CuI thin films, serving as a basis for further quantitative analysis of their chemical compositions were displayed in Supplementary Fig. 3. The results of EDS characterisation conducted on a 3% AZO thin film deposited on an FTO substrate are shown in Supplementary Fig. 3a. The EDS analysis verifies the presence of AZO in the film, as evidenced by distinct peaks corresponding to zinc (Zn), oxygen (O), and aluminium (Al) elements. Specifically, the weight percentage (wt%) of zinc and oxygen in the film is determined to be 39.3% and 26.1%, respectively, which aligns closely with previously reported values[41]. It's worth noting that additional peaks are detected in the EDS spectrum, likely corresponding to elements present in the underlying glass substrate.

In the case of the CuI thin film, the EDS results reveal the presence of peaks corresponding to Cu and I elements (Supplementary Fig. 3b). Notably, the observed element ratio between I/Cu stands at 1.8, consistent with previously reported[29,42,43]. Interestingly, there is a lower atomic percentage of Cu cations relative to anion I, suggesting the existence of Cu vacancies. These vacancies are pivotal in conferring p-type semiconductor properties upon CuI[44]. The EDS analysis also identifies other spectrum peaks, notably silicon, carbon, iodine, and oxygen. The presence of Si is attributed to the substrate, while the occurrence of O and C peaks is likely due to the adsorption of molecules such as $H_2O$, $O_2$, and $CO_2$ on the surface of the CuI crystals when exposed to air[45].

The optical transmittance of AZO and CuI films deposited at various parameters in the 300–800 nm wavelength range is shown in Supplementary Fig. 4. The average transmittance in the visible range for AZO thin films was greater than 75%, a satisfactory threshold for glazing applications. Interestingly, the AZO films demonstrated greater transparency than the ZnO counterparts, though this began to diminish with increased doping, equating to ZnO's transparency at 6% AZO (Supplementary Fig. 4a). This phenomenon can be attributed to the augmentation of doping, which promotes the crystallinity of the film, subsequently diminishing the boundary size and defect density. Such changes elevate transmittance by minimising the reflection of visible light. Furthermore, a decline in AZO transmittance, from 85% at 0.5 h to 75% at 1.5 h, was noted with prolonged electrochemical deposition times due to the film's increasing thickness (Supplementary Fig. 4b). Figure 3e displays the transmission spectrum of the optimized AZO thin film, which exhibits the highest TE performance. The transmission of the optimized AZO thin film reaches up to 76%.

On the other hand, the transmittance of CuI thin films oscillates between 65% and 56%, again a feasible range for glazing applications. Notably, varying annealing temperatures substantially altered transmittance, inducing a drop from 65% to 40% (Supplementary Fig. 4c). Yet,

modifications in NaI concentration seemed to leave the transmittance relatively untouched (Supplementary Fig. 4d). Furthermore, the dipping duration further influenced transmittance where films dipped for 20 and 15 s and demonstrated values between 61% and 56%, while this dropped to 45% for a 5-s dipping time (Supplementary Fig. 4e). This is likely because the longer dipping durations of 20 and 15 s facilitate superior crystallisation and reduced impurities from the restricted deposition time available for copper ions. Finally, increasing the number of cycle counts led to a drop in transmittance from 62% to 56%, attributable to the increased film thickness (Supplementary Fig. 4f). Figure 3e shows the transmission spectrum of the top-performing CuI thin film in terms of TE performance. This film achieves up to 64% transmission.

The energy band gap of the CuI and AZO thin films is derived from the absorption curve using (Eqs. 1 and 2). For the AZO thin film, the optical band gap is ~3.17 eV (Fig. 3f), while for the CuI, it is ~2.8 eV. The band gap for both doped and undoped ZnO ranges between 2.92 and 3.17 eV. Although the typical band gap for an average AZO nanorod stands at 3.5 eV, these samples' observed narrower band gap could be attributed to quantum confinement effects as the diameter is lower than 40 nm[46,47]. Supplementary Fig. 5 illustrates the variation in the band gap of AZO thin films under different aluminium doping amounts. The notable decrease in the band gap for 2% AZO is ascribed to a stress relaxation mechanism, suggesting the introduction of defect states within the band gap[48,49]. In addition, the stress relief in the film, potentially in conjunction with an augmentation in AZO film thickness, could be responsible for the expansion of the 3% AZO bandgap[50]. As a result, there's a pronounced peak in the band gap of the AZO thin film at 3% AZO, signifying the optimal doping concentration for attaining an elevated S.

The band gap for the CuI samples varies between 2.51 and 2.8 eV. Supplementary Fig. 5 illustrates the variation in the band gap of CuI thin films under different annealing temperatures, varying concentrations, and different dipping cycling durations. A distinct trend emerges where the band gap shows an apparent increase with varying NaI concentration, transitioning from 2.51 eV at 0.05 NaI to 2.77 eV at 0.1 NaI concentration, which is attributed to the crystal point defect of copper vacancy. In addition, with an increase in annealing temperature, the band gap arises.

Despite this, a rise in the band gap values is discernible regarding the dipping cycles. Typically, these samples exhibit an appearance like frosted glass. The interference effects observed between 400 and 1500 nm arise from the constructive and destructive interplay between light reflected from the film and the substrate interface. This interaction produces a periodic pattern of peaks corresponding to different wavelengths.

## Thermoelectric performance analysis of the AZO and CuI thin films

A systematic approach was adopted to optimise the S of AZO, involving adjustments to various experimental parameters. These encompassed the aluminium doping concentration, electrochemical deposition duration, and annealing temperature. For a comprehensive analysis, while two parameters were constant, one was variably adjusted to discern its influence on the S. The resultant variations of in-plane S, σ, and PF across different AZO thin film specimens within the ambient temperature range of 275–300 K were shown in Fig. 4a–c.

Figure 4a shows the effect of parameter changes on the Seebeck effect. The samples annealing at a temperature of 500 °C displayed the highest S. Furthermore, the S exhibited an upward trend with increasing film thickness, peaking at 1.5 h of electrochemical depositing. In addition, it is worth noting that doping considerably influenced the S. Specifically, a 3% aluminium doping resulted in a nearly twofold increase compared to standalone sample. However, the S declined after reaching a 6% aluminium doping level.

Both annealing temperatures and film thickness influence σ Fig. 4b. As these parameters escalate, a corresponding rise in σ is observed. Remarkably, Aluminium doping profoundly augments σ, increasing it from values below 10 S.m$^{-1}$ to surpassing 1000 S.m$^{-1}$ at a mere 3% doping level; nonetheless,

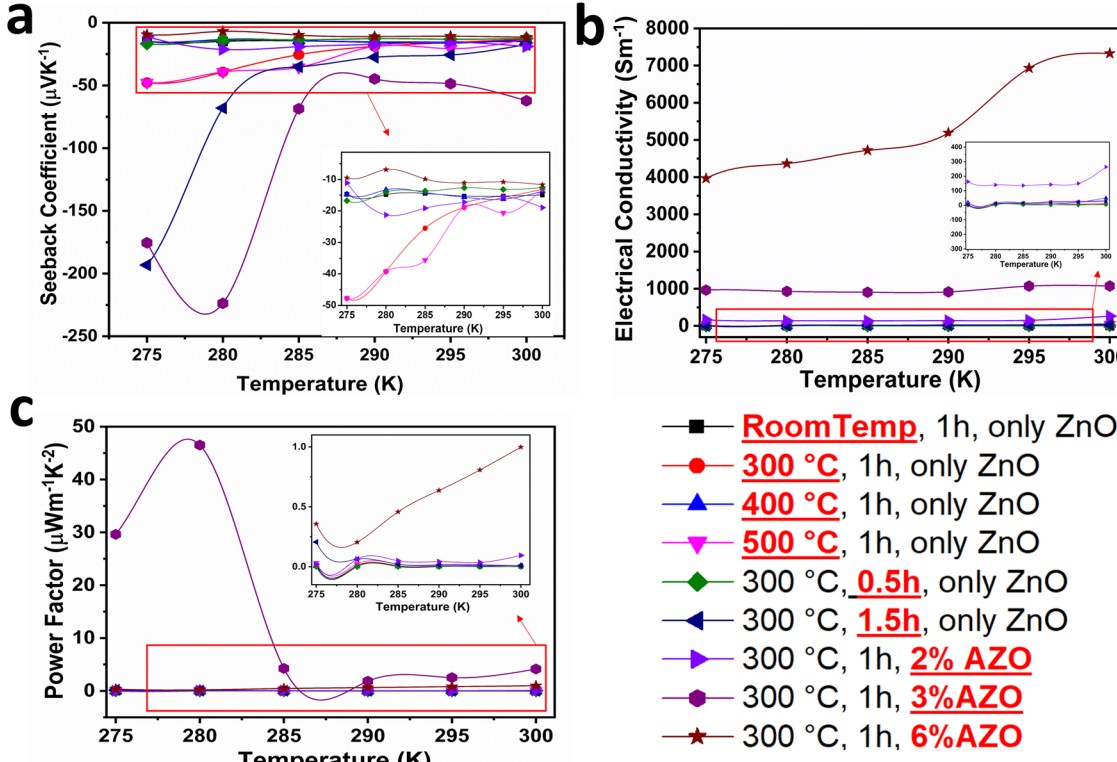

**Fig. 4 | Thermoelectric performance of aluminium-doped zinc oxide (AZO) at different parameters was studied systematically.** Parameters potentially affecting Seebeck were selected. One parameter was tuned (highlighted in red), while others were fixed. **a** The in-plane Seebeck coefficient of AZO for different parameters. **b** The electrical conductivity of AZO for different parameters. **c** The power factor of AZO for different parameters.

with a 6% doping percentage, conductivity spikes to a noteworthy 7500 S.m$^{-1}$.

Figure 4c represents the PF for various samples. The parameters of annealing at 500 °C and a 1.5 h deposition time distinctly modulate PF. However, the influence of doping on PF is substantial, as a 3% doping results in an enhancement of PF by over 40-fold compared to an undoped sample.

A structured strategy was adopted to refine the $S$ of the CuI thin film, tuning parameters such as cationic precursor concentration, immersion duration in the anionic precursor, number of immersion cycles and annealing temperature of the thin films. Initial trials maintained three parameters constant, adjusting one to discern its effect on the $S$. Figure 5a, b, c portrays in-plane $S$, $\sigma$, and PF outcomes across different CuI thin film samples within a temperature spectrum of 275–300 K, respectively. Figure 5a shows the effect of tuning parameter on the in-plane $S$. Notably, a 0.075 NaI cationic precursor concentration for CuI yielded an apex $S$ of 220 μVK$^{-1}$. Altering immersion durations unveiled that a concise 20 s immersion procured optimal results, peaking with a $S$ of 210 μVK$^{-1}$. Further, delving into varied immersion cycles indicated an upward trajectory for the $S$ as cycle counts increased. An apex was achieved with 40 immersion cycles. Moreover, The $S$ burgeons with escalating annealing temperatures, peaking at 150 °C. However, any increment beyond this temperature resulted in a decline.

$\sigma$ analyses across diverse temperatures are depicted in Fig. 5b. A prominent correlation emerged: configurations resulting in heightened $S$ also manifested amplified $\sigma$. Specifically, at a cationic precursor concentration of 0.075 NaI, $\sigma$ reached a commendable 500 S.m$^{-1}$. Analogously, a brief 20 s immersion culminated in an $\sigma$ of 450 S.m$^{-1}$. Annealing further augmented conductivity, peaking at 150 °C. Moreover, cycle count also notably impacted $\sigma$, as evident in its fourfold surge with 40 cycles compared to 35. Figure 5c delineates PF across various CuI thin film samples. Table 1[23,28,29,42–44,51,52] unambiguously portrays that the TE performance of both thin films has substantially evolved from prior studies, which can be attributed to enhanced crystallization, quantum confinement stemming

from reduced particle size in AZO[33,53] and point defects of copper vacancy in CuI[21,37,54].

Figure 6a shows temperature dependence of S, $\sigma$, and PF for the optimised AZO thin film (AZO 3%, annealing at 500 °C, deposited for 1.5 h). At 340 K, the in-plane $S$ reaches −712 μV K$^{-1}$, $\sigma$ is 5230 S.m$^{-1}$ and PF registers 2655 μWm$^{-1}$ K$^{-2}$. Figure 6b shows temperature dependence of S, $\sigma$, and PF for the optimised CuI thin film (0.075 NaI, 20 s immersion time, 40 immersion cycles, 150 °C annealing temperature). At 340 K, the in-plane $S$ reaches 1160 μV K$^{-1}$, $\sigma$ measures 2100 S.m$^{-1}$ and PF registers at 2830 μWm$^{-1}$ K$^{-2}$.

It can be noticed from Fig. 6a, b that there is a simultaneous increase in both S and $\sigma$ with temperature. This behaviour is slightly different from typical trends but has been noted in prior studies[55,56]. This investigation focuses on TEGZ within an operational temperature range of 275–340 K, corresponding to typical building ambient temperatures. The freeze-out region of carrier concentration might falls within this confined temperature range[57]. This specific temperature range is believed to be the reason for the observed trends in Fig. 6a, b. Extending the measurements to 380 K, as shown in supplementary Fig. 6, indicates that beyond 340 K, the trends revert to the more commonly observed behaviour reported in previous reports[58,59]. Supplementary Fig. 6 demonstrates that $\sigma$ increases with temperature, while the Seebeck coefficient increases with temperature up to 350 K for AZO and 340 K for CuI, after which it begins to decrease, thus conforming to the typical trends.

Figure 6c, d illustrates the temperature-dependent $\sigma$ and S of CuI, as influenced FTO substrate. As shown in Fig. 6c, the CuI thin film deposited on FTO exhibits a $\sigma$ value approximately ten times greater than that of the film deposited on bare glass. Similarly, Fig. 6d reveals that the $S$ value of the CuI thin film on FTO glass is six times higher than that of the film on bare glass. In contrast, the $S$ value of FTO glass alone ranges from −19 to −70 μV K$^{-1}$. This significant enhancement in the $S$ value when CuI is deposited on FTO glass highlights the crucial role of the substrate in shaping the TE properties of CuI thin films.

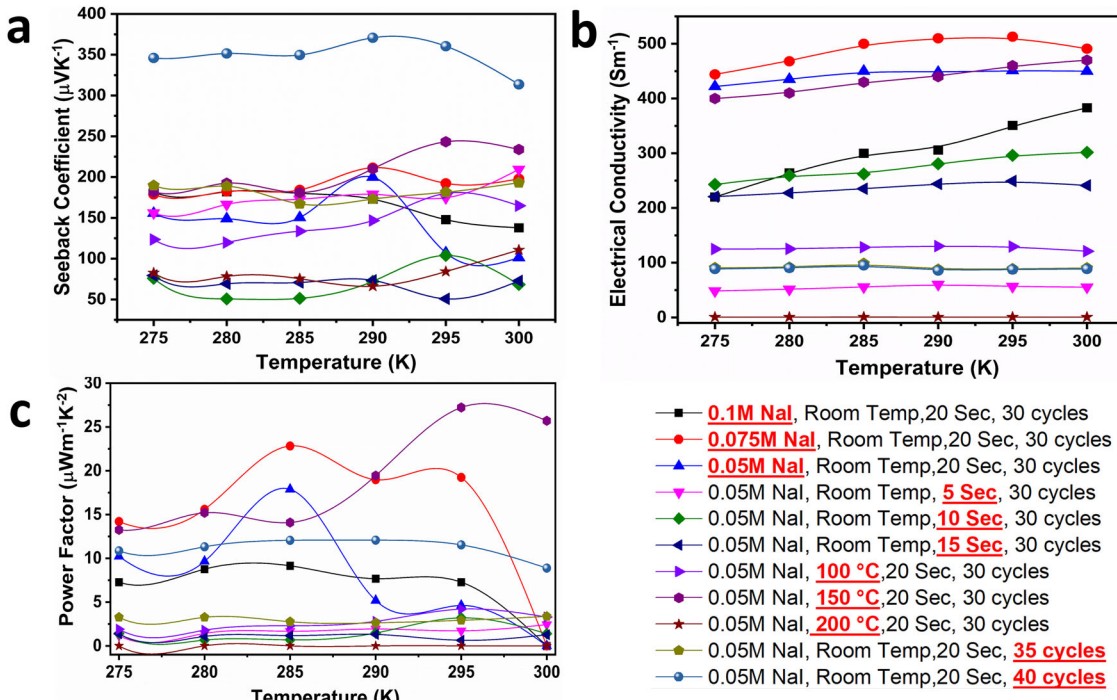

**Fig. 5 | Systematic investigation of thermoelectric performance in CuI across varied parameters.** The thermoelectric performance of CuI was systematically studied under various parameters, focusing on those potentially influencing the Seebeck effect. One parameter was selectively adjusted (highlighted in red and underlined), while the others remained constant. **a** In-plane Seebeck coefficient of CuI under diverse parameter conditions. **b** Electrical conductivity of CuI thin films across different parameter settings. **c** Power factor of CuI thin films under various parameter configurations.

**Table 1 | Comparison of the thermoelectric performance of CuI and AZO achieved in this study and previous works**

| No. | Material | Electrical conductivity (Sm⁻¹) | In plane Seebeck (μV K⁻¹) | Power factor (μWm⁻¹ K⁻²) | Synthesis method | Reference |
|---|---|---|---|---|---|---|
| 1 | CuI | 2100 | 1161 | 2835 | SILAR | This work |
| 2 | CuI | 1000 | 150 | 30 | SILAR | 28 |
| 3 | CuI | 1430 | 207 | 61.2 | SILAR | 29 |
| 4 | CuI | 5000 | 115 | 66.1 | SILAR | 43 |
| 5 | CuI doped Tb | 700 | 550 | 350 | Precipitation method | 42 |
| 6 | CuI | 11,000 | 206 | 470 | Resistive thermal evaporation | 44 |
| 7 | AZO | 5230 | −712 | 2655 | Electrochemical deposition | This work |
| 8 | GZO | – | −62 | – | Resistive thermal evaporation | 44 |
| 9 | Zno:In | 625 | −120 | 9 | SILAR | 29 |
| 10 | Zno:Al | 31,000 | −65 | 130 | Pulsed Laser | 51 |
| 11 | ZnO nanorods | 1100 | −540 | 320 | Chemical bath deposition and microwave methods | 23 |
| 12 | Ti-codoped AZO | 12800 | −102 | 1280 | Metallic buffer layer | 52 |

The purpose of the in-plane S measurement is to optimize TE performance and compare it with previous results as shown in Table 1. However, since this TEGZ application and design requires cross-plane TE properties, the cross-plane S has also been measured, showing that the optimized samples of both AZO and CuI have the highest performance as well. The measured cross-plane S of the optimised AZO, and CuI thin films at 340 K are shown in (Table 2). It can be seen that the cross-plane Seebeck of CuI is lower than the in-plane Seebeck, while in the case of AZO, the cross-plane is higher than the in-plane, and this is due to the anisotropic behaviour of thin films[60]. Moreover, the samples show a low cross-plane k due to phonon scattering in AZO nanorods thin film and point defects of copper vacancy in CuI thin film[37,61–63]. TE materials efficiency could be determined using the Eq. (8) below:

$$ZT = \frac{s^2 \sigma}{k} T \qquad (8)$$

Where k is the thermal conductivity. Impressively, the ZT for both materials is high (Table 2) and could be competitive with the conventional TE materials.

The exceptionally high ZT value reported in Table 2 of our study can be attributed to the significantly enhanced σ, S, and the measured k. In case of AZO, high ZT value is due to enhanced σ as a result of Al doping and

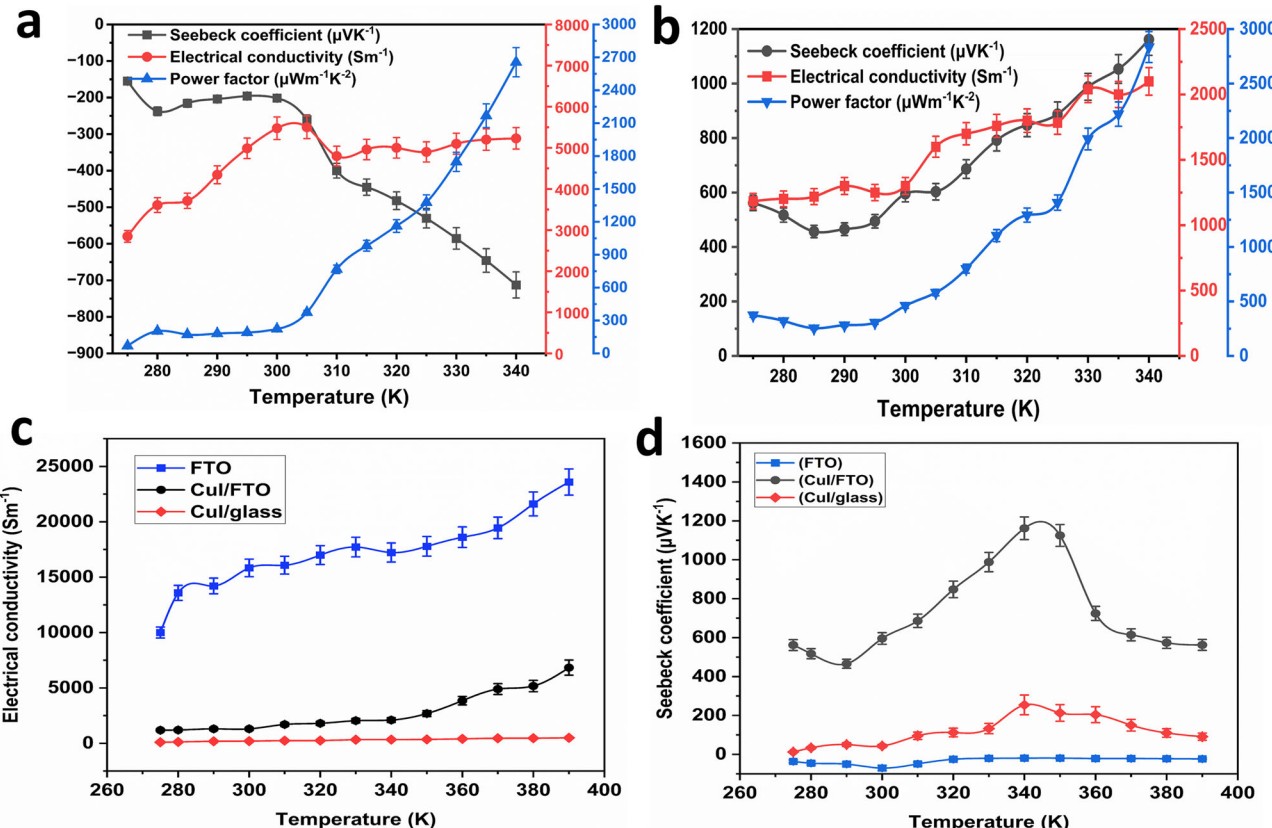

**Fig. 6 | TE properties of optimised sample. a** AZO which has the following parameters 3% aluminium doping, annealing at 500 °C, deposited for 1.5 h. **b** CuI sample which has the following parameters: a cationic precursor concentration of 0.075 NaI, 20-s immersion, annealing at 150 °C, and 40 immersion cycles. The effect of substrate on the thermoelectric performance of CuI illustrates in (**c**) electrical conductivity **d** Seebeck measurements. The error bars of TE properties of optimised sample were determined by standard deviation (SD) from five independent measurements conducted at each temperature point. The SD reflects the variability observed across these trials, indicating the reproducibility of the data. The standard deviation and error bars were calculated using Origin Pro 2024 software.

**Table 2 | Cross-plane TE parameter for optimized samples at 340 K**

| Cross-plane Seebeck coefficient ($\mu V\,K^{-1}$) | Electrical conductivity ($S.m^{-1}$) | Thermal conductivity ($Wm^{-1}K^{-1}$) | Power factor ($\mu Wm^{-1}K^{-2}$) | Figure of merit (ZT) |
|---|---|---|---|---|
| 1077 | 2100 | 1.15 | 2435.851 | 0.72 |
| 950 | 5230 | 1.17 | 4720.075 | 1.37 |

conductivity contribution form FTO substrate whereas nanorod structures boost *S* and reduce $k$[64]. Similarly, for CuI, the high ZT value is also due to high $\sigma$ and *S*, caused by copper vacancy and contribution of FTO substrate[65].

It has been observed that CuI deposited on FTO has a higher $k$ of 1.15 $Wm^{-1}K^{-1}$, compared to 0.85 $Wm^{-1}K^{-1}$ for the thin film deposited on bare glass, which aligns closely with previous studies.[21] Conversely, the $k$ of AZO was found to be lower at 1.17 $Wm^{-1}K^{-1}$ compared to the reported range of 2.2 to 15 $Wm^{-1}K^{-1}$[62]. These observations also indicate a major contribution from the FTO substrate. However, it is notable that the thermal conductivities of CuI and AZO deposited on FTO are less than the measured $k$ of the FTO substrate (1.173 $Wm^{-1}K^{-1}$).

The prototype design of the TEGZ involves a double FTO glazing setup, each measuring 5 × 5 $cm^2$, as illustrated in Fig. 7a. The TEGs were deposited on one of these FTO glazing. These nanogenerators comprise p-type and n-type TE materials on the FTO surface facing the hot side. The FTO itself is used as a conductive electrode. The other FTO glass is pressed against the TEGs on the cold side. This setup allows heat conversion from one side into electricity due to the TE materials. The use of TE materials and FTO also helps to minimise heat losses.

The TEGZ device consists of eight TEGs in total. Connecting electrically in series and thermally in parallel. Each nanogenerator comprises AZO and CuI. In this work, the fabricated nanogenerator was tested, which includes developed AZO and CuI materials deposited on FTO. The operational temperature range falls between 275 and 340 K. To interconnect two nanogenerators, we have employed FTO that pressed against the n-type of one nanogenerator with the p-type of another nanogenerator, as depicted in Fig. 7a.

Figure 7b shows the power output measurement of one nanogenerator according to the measurement device design shown in Fig. 1c. The results show that at 340 K, the temperature difference is around 30 K, the output voltage is around 14 mV, the output current is 2.2 μA, and the output power is 32 nW. To measure the durability of the nanogenerator. The output voltage was measured against time for a month, and each three days, there was one test, as shown in supplementary Fig. 7. It shows a minimum difference in the 30 days, indicating that TEG is reliable and can maintain its performance. However, a scratch has been noticed on the thin films, especially CuI, due to the contact of the golden probe during the measurements, which could be why there was a drop in the output voltage last day.

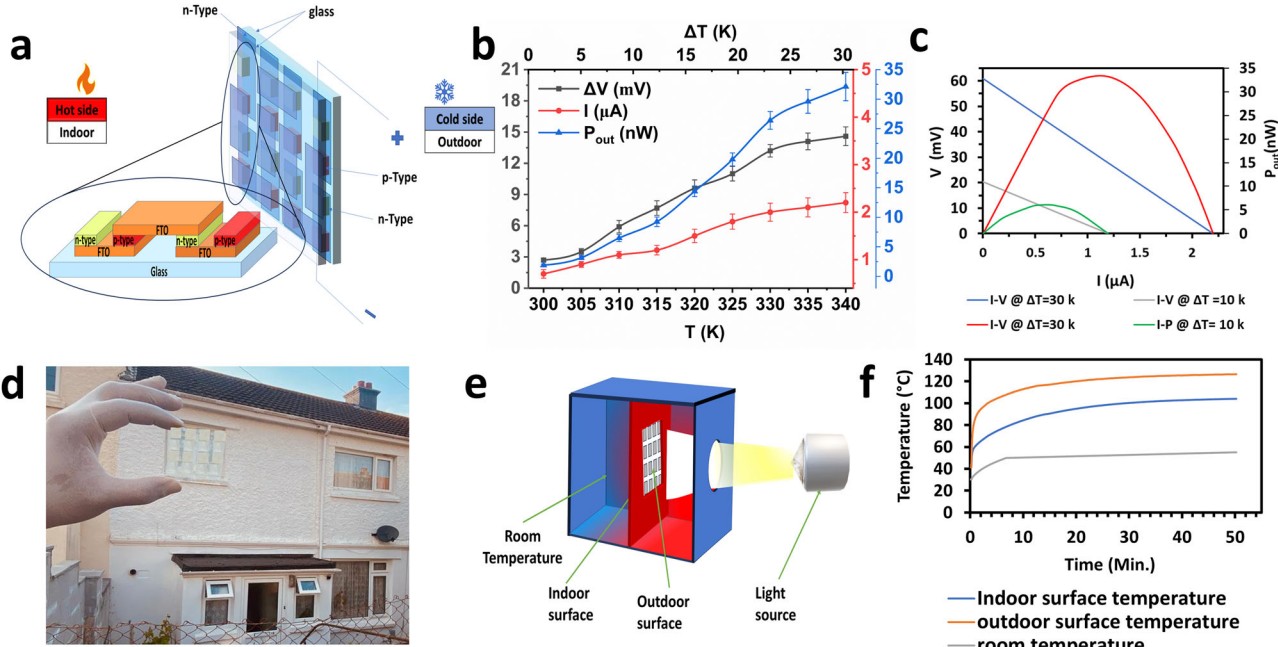

**Fig. 7 | TE glazing performance analysis. a** Illustration of the TE glazing design, which consists of 8 nanogenerators. Each nanogenerator has p-type (CuI) and n-type (AZO) deposited on FTO glass. An FTO glass sandwich from the top connects each of the two nanogenerators. **b** The measuring output of voltage, current and the output power of one nanogenerator. (The error bars of voltage, current and the output power were determined by standard deviation (SD) from five independent measurements conducted at each temperature point. The SD reflects the variability observed across these trials, indicating the reproducibility of the data. The standard deviation and error bars were calculated using Origin Pro 2024 software. **c** The calculated current–voltage (I–V), and power–voltage (P–V) characteristic curves according to the Eqs. 5, 6, 7 at 10 K and 30 K temperature differences. **d** A photograph of TEGZ contains 8 nanogenerators. Each nanogenerator consists of AZO & CuI deposited on FTO glass. **e** A prototype system was used to study the temperature profile of TE glazing. **f** A temperature has been recorded from the fabricated TE glazing ($5 \times 5$ cm²) containing 8 nanogenerators at a middle point of the outside surface, inside and inside the system at room temperature.

Figure 7c illustrates the power output performance, current–voltage (I–V), and power–voltage (P–V) characteristic curves, calculated based on the Eqs. (5–7) for 30 K temperature differences. In the open-circuit condition, the output voltage registered at 60 mV. However, as the output current increased, the voltage decreased linearly. Notably, the observed output power was 33.1 nW, slightly higher than the measured value (32 nW). This discrepancy arises from the variance between the apparent temperature differences applied across the thin-film device and the actual working temperature differences across its thin-film legs. This disparity in temperature is attributed to thermal resistances present at the FTO interface, along with those associated with the top and bottom FTO glass substrates. These factors collectively led to a considerable reduction in the actual temperature gradient across the thin-film legs compared to the apparent $\Delta T$ applied across the entire device.

In Fig. 7d, you can observe a fabricated $5 \times 5$ cm² TEGZ prototype comprising eight TEGs, each consisting of AZO and CuI deposited on FTO. Figure 7e shows a prototype system set up to show the effectiveness of the heat shielding properties[7]. The TEGZ is fixed in the centre of a box made of insulation material. Temperature measurements of the $5 \times 5$ cm² TEGZ were taken using three K-type thermocouples. Dual thermocouples are strategically situated at the core of the TEGZ on opposing sides, quantifying the internal and external surface temperatures. The heat source utilized was a light setup from the WACOM AAA continuous solar simulator, providing an intensity of 1000 Wm⁻² under 1 SUN 1.5 AM conditions. Observed results indicate that the TEGZ decreases the thermal transition from the external interface to the internal one by an impressive margin exceeding 22.5 °C as shown in Fig. 7f. This observation underscores the potential of the TEGZ to induce a minimum of 22.5 °C temperature differential, solely influenced by only one side of the heat. This temperature difference can increase if a cooling system is on the other side. Moreover, the room temperature of the prototype system was raised to 55 °C when the

outside temperature was 120 °C, which means almost half of the heat has faded.

To generate the stipulated amount of electricity using the proposed mechanism, it is imperative to maintain temperature differences between the exterior and interior of a structure, which would typically necessitate the continuous operation of heating or cooling systems. Such an approach, however, may not be environmentally sustainable in the long run. Consequently, while this concept can be a supplementary source to the primary power grid, it is unlikely to replace it as the primary energy source. It is essential to note that consumers are not required to perpetually operate their heating or cooling systems to induce a temperature gradient. The proposed system might find utility primarily when a natural temperature discrepancy exists, catering to the energy demands of smaller appliances. Further economic assessments are imperative to determine the viability of the thermal electric generation zone harnessing naturally occurring temperature gradients. While AZO and CuI are not considered novel materials in the field of materials science, it is important to highlight that their structural and property modifications have the potential to facilitate the design and development of exceptional TE properties.

The implementation of TEGZ technology holds noteworthy promise for creating an energy-positive built environment within urban settings. This assertion is based on several factors can be considered for further study:

(1) Energy harvesting: TEGZ technology utilizes the temperature gradient between indoor and outdoor environments to generate electricity. In urban settings, where buildings are abundant, and energy demand is high, integrating TEGZ panels into building facades, windows, or rooftops can harness substantial amounts of wasted heat energy and convert it into usable electricity.

(2) Reduced dependency on grid power: By generating electricity on-site through TEGZ technology, buildings can reduce their reliance on centralized power grids. This decentralization of energy production

enhances the resilience of urban infrastructure, particularly during peak demand periods or in the event of grid failures or disruptions.

(3) Renewable energy source: TEGZ offer a renewable energy source that can reduce greenhouse gas emissions and mitigate the environmental impact of urban energy consumption.

(4) Urban heat island mitigation: TEGZ panels, when integrated into building exteriors, can help mitigate urban heat islands by reducing the absorption of solar radiation and dissipating excess heat through electricity generation.

(5) Economic viability: Factors such as TEGZ panels' cost-effectiveness, durability, maintenance requirements, and overall lifecycle cost are considered in assessing their impact on the overall energy balance in urban settings.

## Conclusion

Considering growing concerns regarding climate change and the imperative to reduce carbon emissions, TE energy has emerged as a viable solution. This research demonstrates the efficacy of a semi-transparent TEGZ crafted using cost-effective materials through an affordable and scalable fabrication process. Two TE materials, n-type AZO and p-type CuI, were chosen and optimised. Due to its high PF and low $k$, AZO achieved a ZT value of 1.37 at 340 K. These enhancements are linked to reductions in particle size and improved material crystallisation. Concurrently, CuI exhibited a ZT of 0.72 at 340 K due to observed $k$, high $\sigma$ and elevated Seebeck coefficient. The substantial enhancements observed can be attributed to point defects, specifically copper vacancies. The role of FTO glass as a substrate is pivotal in influencing the TE properties of these synthesised materials. Furthermore, the transparency of both films has been optimised, with AZO attaining 76% for an 800 nm thickness and CuI reaching 64% for a 2.7 µm thickness. The TEGZ nanogenerator showcased an output power of 32 nW at 340 K with a temperature gradient around 30 K. In addition, TEGZ's capacity to induce a temperature difference of 22.5 °C between its exterior and interior surfaces accentuates its dual functionality: efficient TE conversion and substantial heat loss reduction. Such modification of fundamental properties in existing materials represents a crucial avenue for achieving performance enhancements.

## Data availability

All data used to create the figures in the main article and supplementary information have been provided in Supplementary Data 1 as source data for the plots. Any additional data is available upon reasonable request to the authors.

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

## Acknowledgements

The authors would like to thank the Engineering and Physical Sciences Research Council (EPSRC), UK, for their support under research grant number EP/T025875/1. However, the EPSRC had no direct involvement in the writing of this article. Mustafa Majid Rashak Al-Fartoos gratefully acknowledges the Higher Committee for Education Development (HCED), Government of Iraq, for funding his PhD studies. For the purpose of open access, the author has applied a 'Creative Commons Attribution (CC BY) licence to any Author Accepted Manuscript version arising from this submission.

## Author contributions

M.M.R.A. Conceptualization, methodology, investigation, data curation, validation, writing-original draft preparation. A.R. Methodology, data curation, validation, writing-review and editing. T.K.M. and A.A.T. Resources, supervision, writing-review and editing, project administration, funding acquisition. All authors discussed the progress of the research and reviewed the manuscript.

## Competing interests

The authors declare no competing interests.
