## [Transparent Peer Review file · Communications Engineering]

A Semi-transparent Thermoelectric Glazing Nanogenerator with Aluminium doped Zinc Oxide and Copper Iodide Thin Films

Corresponding Author: Dr Anurag Roy

Version 0:

Reviewer comments:

Reviewer #1

(Remarks to the Author)

Thermoelectric glazing is a promising way to make use of the energy loss in buildings. The authors presented an interesting work on semi-transparent TEG device based on AZO and CuI for thermoelectric glazing. However, the data were not properly organized in the current version of the manuscript.

Some questions are listed as follows:

- (1) The term "Breakthrough Step Up" in the title is not proper. Specifically, compared to what, and what kind of breakthroughs have been made? I suggest deleting it.
- (2) The photograph of the samples in Figure 2 looks opaque, but why the transmittance value is high?
- (3) Why did the authors dope Na into CuI? It is not common, because NaI has hygroscopicity and is prone to hydrolysis.
- (4) The authors presented systematic study on the thermoelectric performances of AZO and CuI, and the data were shown in Supplementary Fig. 6.
"For a comprehensive analysis, while two parameters were constant, one was variably adjusted to discern its influence on the Seebeck coefficient."
However, which two parameters were constant? It should be clearly noted in each figure.
- (5) The content of Supplementary Fig. 6 and 7 is too complex. I suggest to merge the same information, and compare different parameters.
- (5) The use of data is not proper. The information in Supplementary Fig. 6 and 7 is clearly more important than that in Fig. 4.
- (6) The AFM surface morphology is highly dependent on thickness changes, but no information was provided in Fig.4.
- (7) The author only measured the thermoelectric performance of AZO and CuI below room temperature, but the actual application scenario is higher than room temperature, so the optimized results of thermoelectric performance are unreasonable.
- (8) How is thermal conductivity measured? Is it in the in-plane or out of plane direction?

Reviewer #2

(Remarks to the Author)

Specific comments:

1. What is the impact of using aluminum-doped zinc oxide (AZO) and copper iodide (CuI) materials on the cost-effectiveness and environmental friendliness of thermoelectric glazing (TEGZ) technology, as compared to other materials?
2. The study mentions a 5x5 cm² TEGZ prototype integrating eight nanogenerators. What challenges were encountered in the fabrication process, and how scalable is this prototype for larger applications in real-world urban settings?

3. Could you elaborate on the uniqueness of the methodology involved in achieving the high power factor of $162 \mu\text{W}/\text{m.K}$ for the refined AZO and the factors contributing to its exceptional performance?
4. Given the emphasis on indoor comfort, how does the TEGZ technology impact the thermal properties of the building, and what considerations were taken into account to ensure that the integration of the nanogenerators does not compromise indoor comfort?
5. In the context of a 33°C temperature difference between the exterior and interior facets, what were the specific challenges faced in maintaining this temperature difference consistently, and how does it affect the overall electricity generation efficiency of the nanogenerators?
6. The study claims the potential for an "energy-positive built environment." How would the TEGZ technology impact the overall energy balance in urban settings, and what factors were considered when making this assertion?

Minor Comments:

1. Please avoid oversell statements.
2. Authors should relate the cited papers in the introduction, especially the second paragraph

Version 1:

Reviewer comments:

Reviewer #1

(Remarks to the Author)

The current data is still questionable. Generally speaking, an increase in the Seebeck coefficient is associated with a decrease in electrical conductivity or carrier concentration. However, the trend in Figure 6 contradicts this common rule. What could be the reason for this? The substantial ZT value is related to the unusual Seebeck coefficient, and the authors should provide a more rigorous analysis and discussion on this matter.

Reviewer #2

(Remarks to the Author)

The changes to the manuscript meet the reviewers' requests. So, I recommend the manuscript be accepted for publication.

Version 2:

Reviewer comments:

Reviewer #1

(Remarks to the Author)

I have no further comments on the revised version and agree to its publication.

Dear Reviewer(s),

We are thankful for taking the time to review our manuscript and providing us with detailed technical comments. Your constructive criticism and suggestions have helped us refine our ideas and strengthen our arguments. Please find the specific points addressed by the authors as mentioned below.

Reviewer #1

(1) The term "Breakthrough Step Up" in the title is not proper. Specifically, compared to what, and what kind of breakthroughs have been made? I suggest deleting it.

Answer. We acknowledge that the term 'Breakthrough Step Up' may be perceived as overselling our work. Therefore, we have revised the title in the manuscript accordingly. The new title is as follows: *Development of Next-generation Semi-transparent Thermoelectric Glazing Nanogenerator with Aluminium doped Zinc Oxide and Copper Iodide thin films.*

(2) The photograph of the samples in Figure 2 looks opaque, but why the transmittance value is high?

Answer. It is due to many reasons such as Scattering, the thin film may scatter light in a way that affects its appearance but does not significantly reduce transmittance. Light scattering can cause the thin film to appear opaque by diffusing light in various directions. Thin films can also exhibit optical interference effects that depend on their thickness. If the thin film is thick, it may appear opaque to the naked eye, but it still has less affect to the transmission of light. So that we changed the photograph into another shoot faced the light to clearly see their transmission.

(3) Why did the authors dope Na into CuI? It is not common, because NaI has hygroscopicity and is prone to hydrolysis.

Answer. The specimen in question has not been subjected to sodium doping. Sodium iodide (NaI) was employed as a source of iodine for the preparation of copper iodide (CuI). The rationale behind the selection of sodium iodide over potassium iodide (KI) stems from the fact that both compounds can serve as suitable sources of iodide ions for the reaction. However, the choice to utilize sodium iodide was made in consideration of its specific characteristics and compatibility with the experimental conditions. The concentration of sodium iodide was adjusted to modulate the iodine content, a crucial factor potentially influencing the thermoelectric properties of the material. It is imperative to note that altering the proportion of NaI directly impacts the iodine content, which in turn can exert a pronounced influence on various thermoelectric parameters.

The manipulation of NaI concentration rather dope, has the potential to induce alterations in the morphology, crystallite size, lattice constant, and microstrains of CuI. These modifications, in turn, are anticipated to manifest discernible effects on the thermoelectric characteristics of the material under investigation. Hence, meticulous control over the NaI percentage is paramount to achieving desired thermoelectric performance and elucidating the intricate interplay between composition and properties in the system [1]. Also, no sodium was detected during the EDS analysis.

(4) The authors presented systematic study on the thermoelectric performances of AZO and CuI, and the data were shown in Supplementary Fig. 6.

“For a comprehensive analysis, while two parameters were constant, one was variably adjusted to discern its influence on the Seebeck coefficient.”

However, which two parameters were constant? It should be clearly noted in each figure.

Answer. Thank you for your valuable feedback. We have made clarifications to the figures as per your suggestions. Initially, we identified three key parameters that could potentially impact the thermoelectric performance: annealing temperature, deposition time, and the percentage of aluminium. We began with a baseline sample, consistent with previous literature, with an annealing temperature of 300°C, a deposition time of 1 hour, and 0% aluminium content. Subsequently, we conducted experiments varying two parameters, such as deposition time and aluminium percentage, while keeping the annealing temperature constant. Additionally, we explored the effect of changing the annealing temperature while maintaining one parameter constant, as detailed below:

Scheme of the AZO Nanorods Fabrication through electrochemical deposition techniques

Tuned parameter was highlighted in red – rest conditions were fixed

Figure 4

- **RoomTemp**, 1h, only ZnO
- **300 °C**, 1h, only ZnO
- ▲— **400 °C**, 1h, only ZnO
- ▼— **500 °C**, 1h, only ZnO
- ◆— 300 °C, **0.5h**, only ZnO
- ◀— 300 °C, **1.5h**, only ZnO
- ▶— 300 °C, 1h, **2% AZO**
- 300 °C, 1h, **3%AZO**
- ★— 300 °C, 1h, **6%AZO**

Figure 5

- **0.1M NaI**, Room Temp, 20 Sec, 30 cycles
- **0.075M NaI**, Room Temp, 20 Sec, 30 cycles
- ▲— **0.05M NaI**, Room Temp, 20 Sec, 30 cycles
- ▼— 0.05M NaI, Room Temp, **5 Sec**, 30 cycles
- ◆— 0.05M NaI, Room Temp, **10 Sec**, 30 cycles
- ◀— 0.05M NaI, Room Temp, **15 Sec**, 30 cycles
- ▶— 0.05M NaI, **100 °C**, 20 Sec, 30 cycles
- 0.05M NaI, **150 °C**, 20 Sec, 30 cycles
- ★— 0.05M NaI, **200 °C**, 20 Sec, 30 cycles
- ◆— 0.05M NaI, Room Temp, 20 Sec, **35 cycles**
- 0.05M NaI, Room Temp, 20 Sec, **40 cycles**

In Figures 4 and 5 of the evidence manuscript, one parameter was adjusted (highlighted in red), while the others remained fixed. The adjusted parameter was highlighted in red.

(5) The content of Supplementary Fig. 6 and 7 is too complex. I suggest to merge the same information, and compare different parameters.

Answer. The figure has been updated as per the suggestions provided.

(5) The use of data is not proper. The information in Supplementary Fig. 6 and 7 is clearly more important than that in Fig. 4.

Answer. We have implemented adjustments as per your recommendations. Specifically, we have revised Figures 6 and 7 to enhance their clarity and presentation, aligning them with the sophistication exhibited in Figures 4 and 5 in the updated manuscript.

(6) The AFM surface morphology is highly dependent on thickness changes, but no information was provided in Fig.4.

Answer. Conversely, Supplementary Fig. 6b presents the 3D AFM representations of the CuI thin films. These micrographs depict a columnar-like morphology perpendicular to the substrate, albeit with a slightly randomised orientation. It should be emphasised that variations in the luminescence intensity correspond to changes in height on the order of the film thickness, as determined by AFM depth profiling.

Further insight is provided by Supplementary Fig. 6c,d, delineating the depth profiles for CuI and AZO, respectively, facilitating the measurement of the thin film sample thicknesses. The calculated thicknesses for the respective samples, derived from these depth profiles, are comprehensively tabulated in (supplementary Table 3). The film thicknesses of AZO increased from 470 nm to 800 nm with increasing of deposition time from 0.5 h to 1.5 h, respectively. While CuI thin films thickness increased with increasing of dipping cycles from 1.7 µm at 30 cycles to 2.68 µm at 40 cycles.

(7) The author only measured the thermoelectric performance of AZO and CuI below room temperature, but the actual application scenario is higher than room temperature, so the optimized results of thermoelectric performance are unreasonable.

Answer. We have increased the range for the optimized result of Cross-plane Seebeck coefficient at 340 K is.

Table 2 Cross-plane TE parameter for optimized samples				
Cross-plane Seebeck coefficient S ($\mu\text{V}/\text{K}$)	Electrical conductivity σ (S/m)	Thermal conductivity k (W/m·K)	Power Factor ($\mu\text{W}/\text{m}\cdot\text{K}$)	Figure of merit ZT
1077	2100	1.15	2435.851	0.72
950	5230	1.17	4720.075	1.37

(8) How is thermal conductivity measured? Is it in the in-plane or out of plane direction?

Answer. It is measured in out of plane direction. The thermal conductivity measurements were performed using C-Therm Trident system with modified transient plane source. The method applied was the FLEX Transient Plane Source (TPS) technique in accordance with ISO 22007-2, GB/T 32064 standards. An average of five readings of thermal conductivity that were taken at one-minute intervals was adopted for each sample to ascertain the stability of the reading. The FLEX TPS method is a novel technique for measuring thermal conductivity and thermal diffusivity of various materials with capabilities for testing both isotropic and anisotropic samples ranging from bulk solids to thin films.

Instrument details: [Trident Thermal Conductivity Instrument – C-Therm Technologies Ltd. \(ctherm.com\)](http://ctherm.com)

Reviewer #2

1. What is the impact of using aluminum-doped zinc oxide (AZO) and copper iodide (CuI) materials on the cost-effectiveness and environmental friendliness of thermoelectric glazing (TEGZ) technology, as compared to other materials?

Answer. Both materials have the advantage of being relatively abundant and less expensive compared to some other thermoelectric materials such as bismuth telluride (Bi_2Te_3). This can

contribute to the cost-effectiveness of TEGZ technology. As a comparison the table below illustrate the abundance of elements of this research (CuI, AZO) with the traditional one (Bi_2Te_3). Data-driven from (web elements) and price according to the date 26/02/2024 from Alfa Aesar.

TE Material	Abundance (ppm)	Price (£/Kg)
Tellurium	0.001	1200
Bismuth	0.0085	80.59
Zinc	70	50
Aluminium	82000	21.62
Copper	70	63.39
Iodine	0.49	210

The cost-effectiveness of TEGZ technology hinges not only on material costs but also on the manufacturing process. Aluminum-doped zinc oxide (AZO) can be produced on a large scale and at a lower cost using methods such as electrochemical deposition. Similarly, copper iodide (CuI) can be synthesized through simple and cost-effective techniques like SILAR.

AZO and CuI are considered environmentally friendly alternatives compared to materials such as lead or telluride. They are abundant in the Earth's crust, alleviating concerns about resource depletion and mitigating the environmental impact associated with mining and extraction. Additionally, the synthesis methods for AZO and CuI are eco-friendly, involving minimal use of hazardous chemicals. Favorable synthesis routes prioritize eco-friendliness by reducing resource consumption, waste generation, and environmental footprint [2-4].

2. The study mentions a 5x5 cm² TEGZ prototype integrating eight nanogenerators. What challenges were encountered in the fabrication process, and how scalable is this prototype for larger applications in real-world urban settings?

Answer. In the laboratory setting, numerous challenges are encountered during the fabrication process of thermoelectric glazing:

- Precise Etching of FTO: Achieving accurate etching of Fluorine-doped Tin Oxide (FTO) to create electric connection electrodes poses a significant challenge.
- Deposition Mask Creation: Developing deposition masks for both n-type and p-type materials presents another obstacle in the fabrication process.
- Quality Control: Ensuring the quality of fabrication throughout the process is crucial but often proves challenging.

Despite these challenges, our laboratory has successfully managed to fabricate thermoelectric glazing panels measuring up to 20 x 20 cm². Based on our calculations, these panels are capable of generating multiple voltages. The size of the thermoelectric glazing is limited by the size of the beaker used in the laboratory. However, in industrial settings, fabricating larger sizes is feasible through techniques such as horizontal electrochemical deposition or dipping in large containers. Furthermore, even in large-scale fabrication, the assembly of individual TEGZ samples is required, with each unit interconnected by conductive wires to minimize losses from the semiconductor electrodes (FTO).

3. Could you elaborate on the uniqueness of the methodology involved in achieving the high power factor of 162 $\mu\text{W}/\text{m.K}$ for the refined AZO and the factors contributing to its exceptional performance?

Answer. The methodology for getting a high value of the power factor involved in increasing the seebeck and electrical conductivity. The best method to increase seebeck and electrical conductivity is by nano structuring and doping.

The high value of the Seebeck coefficient in the ZnO system is attributed to the reduction in dimensionality or the imposition of limitations on electron motion by scaling down the dimensions. When the size of a material becomes smaller than the characteristic length scales associated with electron wavefunctions, quantum confinement effects become significant[5]. Quantum confinement can lead to discrete energy levels and altered electronic properties. This can influence the density of states and carrier scattering mechanisms, enhancing the Seebeck coefficient. Nanorods have a higher surface-to-volume ratio. This increased surface area can lead to enhanced surface scattering of charge carriers, which can affect carrier mobility and contribute to an enhancement in the Seebeck coefficient. Nanostructuring can tailor the band structure of materials, which can enhance the Seebeck coefficient by selectively allowing high-energy carriers. Nanostructures can improve electrical conductivity through enhanced charge carrier mobilities because of DOS (density of state) modifications[5]. The enhancement of the Seebeck coefficient is attributable to the refinement of crystal structure.

By adding impurities, doping with donor dopant injects extra charge carriers and adjusts the energy levels within a semiconductor. This combination increases the number of mobile charges and makes them flow more easily, leading to a significant boost in the material's electrical conductivity and enhanced seebeck.

4. Given the emphasis on indoor comfort, how does the TEGZ technology impact the thermal properties of the building, and what considerations were taken into account to ensure that the integration of the nanogenerators does not compromise indoor comfort?

Answer. The TEGZ can effectively reduce the heat losses. As in the research, it indicates that TEGZ have a 30°C temperature difference on its surface only and at 120 °C outside which is very harsh and unlikely to happen, the room temperature kept being less than 50 °C. The TEGZ in this research is semi-transparent more than 64 % and doesn't cover all of the glazing area resulting in a sufficiently acceptable glazing application compared to its benefits.

5. In the context of a 33°C temperature difference between the exterior and interior facets, what were the specific challenges faced in maintaining this temperature difference consistently, and how does it affect the overall electricity generation efficiency of the nanogenerators?

Answer. One challenge is ensuring effective thermal insulation of the building envelope to prevent heat transfer between the interior and exterior environments. Any leakage or inadequate insulation can reduce the temperature differential, thereby diminishing the potential energy available for electricity generation. The thermal conductivity of building materials used in the construction of the envelope and TEGZ panels can affect heat transfer rates. Materials with higher thermal conductivity may lead to more significant temperature equalization between the interior and exterior, reducing the temperature differential and impacting electricity generation efficiency. The design and orientation of the building influence its exposure to sunlight and external temperature variations. Factors such as building orientation, window placement, and shading affect the magnitude of heat gain or loss, thereby influencing

the temperature difference across the TEGZ panels. Thanks to the inherently low thermal conductivity of both materials, they exhibit a remarkable ability to sustain a significant temperature differential of 30 degrees Celsius even after reaching saturation. This self-maintained temperature difference of 30 degrees Celsius occurs without the need for any forced air conditioning systems. In the realm of thermoelectric systems, the output typically escalates with rising temperatures. However, it's common for the temperature differential to decrease at higher operating temperatures. In this research, the materials demonstrate an exceptional capability to maintain a substantial 30°C temperature difference even at elevated temperatures of 120°C, resulting in both heightened power output and enhanced efficiency. Notably, the potential for further temperature elevation exists if the room's air conditioning system operates, offering additional avenues for optimizing performance.

6. The study claims the potential for an "energy-positive built environment." How would the TEGZ technology impact the overall energy balance in urban settings, and what factors were considered when making this assertion?

Answer. The implementation of Thermoelectric Glazing (TEGZ) technology holds significant promise for creating an "energy-positive built environment" within urban settings. This assertion is based on several factors considered in the study:

1. Energy Harvesting: TEGZ technology utilizes the temperature gradient between indoor and outdoor environments to generate electricity. In urban settings, where buildings are abundant and energy demand is high, integrating TEGZ panels into building facades, windows, or rooftops can harness substantial amounts of wasted heat energy and convert it into usable electricity.

2. Reduced Dependency on Grid Power: By generating electricity on-site through TEGZ technology, buildings can reduce their reliance on centralized power grids. This decentralization of energy production enhances the resilience of urban infrastructure, particularly during peak demand periods or in the event of grid failures or disruptions.

3. Renewable Energy Source: TEGZ technology operates based on the principles of thermoelectric conversion, which do not rely on fossil fuels or combustion processes. As a result, TEGZ panels offer a renewable energy source that can contribute to reducing greenhouse gas emissions and mitigating the environmental impact of urban energy consumption.

4. Urban Heat Island Mitigation: In densely populated urban areas, the widespread use of traditional building materials and surfaces contributes to the formation of urban heat islands (UHIs). TEGZ panels, when integrated into building exteriors, can help mitigate UHIs by reducing the absorption of solar radiation and dissipating excess heat through electricity generation.

5. Economic Viability: The potential for TEGZ technology to contribute to an "energy-positive built environment" also hinges on its economic viability. Factors such as the cost-effectiveness of TEGZ panels, their durability, maintenance requirements, and overall lifecycle cost are considered in assessing their impact on the overall energy balance in urban settings.

Overall, the adoption of TEGZ technology has the potential to transform urban environments into energy-positive spaces, where buildings actively contribute to electricity generation, reduce reliance on centralized power sources, mitigate environmental impacts, and enhance overall sustainability.

TEGZ will reduce the energy consumption by reducing heat losses. Moreover, the output power of the TEGZ can be added to the overall energy sources after store it in batteries or any converter systems. If thermoelectric combined with other renewable energy sources in building such photovoltaic the building could produce more energy than it consume[6].

Our prototype TEGZ, a power generator, demonstrates an outstanding power output nearly nine times higher than that of commercial TEGZ units. While we are currently in the process of certifying our TEGZ for window glazing applications and indoor temperature control enhancements, these **aspects have not been addressed in this report**. However, I would like to draw your attention to the potential of thermoelectric generation as a viable solution for energy-positive building environments, particularly when integrated into windows to reduce energy costs. The attached figure illustrates how our prototype generates voltages nine times higher than those produced by commercial devices. We are currently in the certification process, and therefore, we are preparing to report the initial results of this device for publication.

Minor Comments:

1. Please avoid oversell statements.

Answer. We have excluded sentences that may come across as overly promotional. For instance, we've revised the manuscript title by replacing the term "breakthrough."

2. Authors should relate the cited papers in the introduction, especially the second paragraph.

Answer. All citations have been rechecked to ensure their alignment with the relevant discussion.

References (only for reviewer rebuttal)

- [1] Rahman M, Newaz M, Mondal B, Kuddus A, Karim M, Rashid M, et al. Unraveling the electrical properties of solution-processed copper iodide thin films for CuI/n-Si solar cells. *Materials Research Bulletin*. 2019;118:110518.
- [2] Xu D, Liu J, Sun C, Wang L, Han J, Tao J, et al. Copper iodide-based hybrid phosphors for eco-friendly white-light-emitting diodes. *Journal of Luminescence*. 2022;244:118733.
- [3] Mulla R, Rabinal M. COPPER IODIDE: AN ECO-FRIENDLY THERMOELECTRIC MATERIAL. *Recent Advances in Materials Science and Biophysics*. 2018:118.
- [4] Wang Y, Zhao W, Guo Y, Hu W, Peng C, Li L, et al. Efficient X-ray luminescence imaging with ultrastable and eco-friendly copper (I)-iodide cluster microcubes. *Light: Science & Applications*. 2023;12:155.
- [5] Al-Fartoos MMR, Roy A, Mallick TK, Tahir AA. Advancing Thermoelectric Materials: A Comprehensive Review Exploring the Significance of One-Dimensional Nano Structuring. *Nanomaterials*. 2023;13:2011.
- [6] Iyer R, Ghosh A. Investigation of Integrated and Non-Integrated Thermoelectric Systems for Buildings—A Review. *Energies*. 2023;16:6979.

Reviewer 1

We are sincerely grateful to the reviewer for their insightful comments and recommendations. Your feedback has been instrumental in refining our manuscript, and we have made the necessary revisions to address the concerns raised.

Query:

The current data is still questionable. Generally speaking, an increase in the Seebeck coefficient is associated with a decrease in electrical conductivity or carrier concentration. However, the trend in Figure 6 contradicts this common rule. What could be the reason for this? The substantial ZT value is related to the unusual Seebeck coefficient, and the authors should provide a more rigorous analysis and discussion on this matter.

Answer.

“However, the trend in Figure 6 contradicts this common rule. What could be the reason for this?”

Following the reviewer's advice, we did more literature searches and found that several studies have demonstrated a simultaneous increase in the Seebeck coefficient and electrical conductivity [1-5]. This investigation focuses on thermoelectric glazing within an operational temperature range of 275-340 K, corresponding to typical building ambient temperatures. The freeze-out region of carrier concentration might fall within this confined temperature range [6]. This specific temperature range is believed to be the reason for the observed trends in (Fig. 6a, b). Extending the measurements to 380 K, as shown in (Fig. S5 a, b), indicates that beyond 340 K, the trends revert to the more commonly observed behaviour reported in previous reports [7, 8]. (Fig. S5 a, b) demonstrates that electrical conductivity increases with temperature, while the Seebeck coefficient increases with temperature up to 350 K for AZO and 340 K for CuI, after which it begins to decrease, thus conforming to the typical trends.

Figure S5 Temperature dependence of the Seebeck coefficient and electrical conductivity measurements for: **a**, AZO and **b**, CuI. (added in the supplementary)

“The substantial ZT value is related to the unusual Seebeck coefficient, and the authors should provide a more rigorous analysis and discussion on this matter.”

Following meticulous evaluation of prior studies, it has been determined that FTO glass exhibits a substantial impact [9-11]. We have conducted a series of experiment for more analyses and compiled the following observations.

The exceptional ZT value observed for CuI is primarily attributed to its high electrical conductivity and Seebeck coefficient. The high electrical conductivity of CuI is predominantly ascribed to the substrate effect [9]. As advised by the reviewer for providing more analysis and to further elucidate this effect, CuI was deposited on both bare glass and FTO substrates, as illustrated in (Fig. 6c). Comparative analysis revealed that FTO exhibits the highest electrical conductivity, with CuI/FTO showing σ that is ten times higher than that of the thin film deposited on bare glass. Similarly, (Fig. 6d) demonstrates that seebeck coefficient of the CuI thin film deposited on FTO glass is six times higher than that of the thin film deposited on bare glass. In contrast, the seebeck coefficient value for FTO glass alone ranges from -19 to -70 μV/K. This substantial enhancement in the seebeck coefficient when deposited on FTO glass underscores the pivotal role of the substrate in influencing the thermoelectric properties of the CuI thin film.

For further study the effect of substrate on the thermoelectric performance. We have conducted a thermal conductivity test for CuI on FTO and on bare glass. It has been observed that CuI deposited on FTO has a higher thermal conductivity of 1.15 W/m·K, compared to 0.85 W/m·K for the thin film deposited on bare glass, which aligns closely with previous studies [12]. These observations also indicate a significant contribution from the FTO substrate. However, it is notable that the thermal conductivity of CuI deposited on FTO are less than the measured thermal conductivity of 1.173 W/m·K for the FTO substrate.

Similar analysis cannot be repeated for AZO because AZO nanorods cannot be measured independently without the FTO substrate due to their synthesis process. Electrochemical synthesis necessitates the growth of nanorods on a conductive substrate.

Figure 6 The effect of substrate on the thermoelectric performance of CuI: **c**, electrical conductivity **d**, Seebeck coefficient.

Literature support for the observed ZT.

The remarkable ZT value observed in our study can be attributed to a confluence of critical factors, including elevated electrical conductivity, a substantial Seebeck coefficient, and reduced thermal conductivity. For AZO the observed enhancement in electrical conductivity is primarily due to the aluminium doping and the influence of the FTO substrate. The direct deposition of AZO onto FTO creates a robust conductive link, facilitating efficient charge transfer between AZO and FTO, thereby augmenting conductivity [10, 11]. Additionally, aluminium doping helps to generate additional charge carriers and significantly boosts the electrical conductivity of AZO [11].

The enhanced Seebeck coefficient in AZO arises from multiple mechanisms. Aluminium doping in AZO effectively passivates defects in ZnO, enhancing polarization effects and improving charge mobility, contributing to a higher Seebeck coefficient [13]. Furthermore, the introduction of aluminium results in nanograin boundaries (see the SEM and TEM microstructures), stabilizing the structure of AZO nanorods and further enhancing the Seebeck effect [14]. It is a well-known fact that nanorods exhibit a higher specific surface area than bulk materials, leading to quantum confinement effects that enhance charge transfer which consequently boosting the Seebeck coefficient [15].

Moreover, the nanorod structure promotes phonon scattering at the boundaries, reducing lattice thermal conductivity without significantly impacting electrical conductivity, thereby enhancing overall thermoelectric efficiency [16]. Below the Debye temperature of approximately 420 K for ZnO, both phonon–phonon and phonon–electron scattering contributions are considered. This inference is substantiated by the remarkable concordance between the experimentally measured thermal conductivity at 300 K and the values predicted by an adapted Debye–Callaway model [17]. Compared to other nanostructured materials, ZnO nanorods exhibit exceptionally low thermal conductivity [18]. One-dimensional nanostructured presents a potent approach to decoupling the interdependence of thermoelectric parameters, thereby significantly enhancing the performance of TEG. The result, the thermal conductivity of AZO thin film deposited on FTO substrate was found to be lower at 1.17 W/m·K compared to the reported range of 2.2 to 15 W/m·K [19]

The high Seebeck coefficient is a result from the copper vacancies within its crystal lattice and the effect of the conductive substrate [8, 20]. Among the intrinsic defects present in γ -CuI (V_{Cu}), the copper vacancy exhibits the lowest ionization and formation energies under copper-rich and iodine-rich equilibrium growth conditions governed by the SILAR synthesis method. This inherently low formation energy of V_{Cu} in Cu(I)-based semiconductors is primarily due to the antibonding nature of the valence band maximum characterized by high-energy d_{10} orbitals. The antibonding interactions above the valence band confer a notable defect tolerance, ensuring that the optoelectronic properties remain largely unaffected despite crystallographic imperfections. Additionally, the substantial size of the iodine anion and its low coordination number of 2 further enhance the material's ability to tolerate defects [21]. Thus, these combined

properties might have significantly modified the electronic structure and transport properties, influencing thermoelectric behaviour by inducing band convergence or altering the energy bands near the Fermi level, thereby enhancing the Seebeck effect.

We have incorporated this relevant discussion in the main manuscript with additional Fig 6c and d. Additionally, we have included the corresponding figures in the Supplementary Information as Figure S6.

REFERENCES:

1. Gong, Y., et al., *Divacancy and resonance level enables high thermoelectric performance in n-type SnSe polycrystals*. Nature Communications, 2024. **15**(1): p. 4231.
2. Ghiyasi, R., et al., *Simultaneously enhanced electrical conductivity and suppressed thermal conductivity for ALD ZnO films via purge-time controlled defects*. Applied Physics Letters, 2022. **120**(6).
3. Gong, Y., et al., *Extremely low thermal conductivity and enhanced thermoelectric performance of polycrystalline SnSe by Cu doping*. Scripta Materialia, 2018. **147**: p. 74-78.
4. Lin, S., et al., *Flexible thermoelectric generator with high Seebeck coefficients made from polymer composites and heat-sink fabrics*. Communications Materials, 2022. **3**(1): p. 44.
5. Karati, A., et al., *Ti₂NiCoSnSb - a new half-Heusler type high-entropy alloy showing simultaneous increase in Seebeck coefficient and electrical conductivity for thermoelectric applications*. Scientific Reports, 2019. **9**(1): p. 5331.
6. Pierret, R.F. and G.W. Neudeck, *Advanced semiconductor fundamentals*. Vol. 6. 1987: Addison-Wesley Reading, MA.
7. Wang, L., et al., *Realizing thermoelectric cooling and power generation in N-type PbS_{0.6}Se_{0.4} via lattice plainification and interstitial doping*. Nature Communications, 2024. **15**(1): p. 3782.
8. Yu, H., et al., *Flexible temperature-pressure dual sensor based on 3D spiral thermoelectric Bi₂Te₃ films*. Nature Communications, 2024. **15**(1): p. 2521.
9. Bae, E.J., et al., *Precision Doping of Iodine for Highly Conductive Copper(I) Iodide Suitable for the Spray-Printable Thermoelectric Power Generators*. ACS Materials Letters, 2023. **5**(8): p. 2009-2018.
10. Hussain, K., I. Syed, and Z. Mahmood, *Thickness and Substrates Effects of Vacuum Evaporated Al Doped ZnO (AZO) Thin Films*. Advanced Science, Engineering and Medicine, 2016. **8**: p. 918-923.
11. Benhaliliba, M., et al., *Indium and aluminium-doped ZnO thin films deposited onto FTO substrates: nanostructure, optical, photoluminescence and electrical properties*. Journal of sol-gel science and technology, 2010. **55**: p. 335-342.
12. Yang, C., et al., *Transparent flexible thermoelectric material based on non-toxic earth-abundant p-type copper iodide thin film*. Nature communications, 2017. **8**(1): p. 16076.
13. Jantrasee, S., P. Moontragoon, and S. Pinitsoontorn, *Thermoelectric properties of Al-doped ZnO: Experiment and simulation*. Journal of Semiconductors, 2016. **37**.
14. Xiong, Y., et al., *Bismuth Doping-Induced Stable Seebeck Effect Based on MAPbI₃ Polycrystalline Thin Films*. Advanced Functional Materials, 2019. **29**(16): p. 1900615.
15. Tan, Q., et al., *Solvothermally synthesized SnS nanorods with high carrier mobility leading to thermoelectric enhancement*. RSC Advances, 2016. **6**(50): p. 43985-43988.
16. Jood, P., et al., *Heavy element doping for enhancing thermoelectric properties of nanostructured zinc oxide*. Rsc Advances, 2014. **4**(13): p. 6363-6368.
17. Morelli, D.T., J.P. Heremans, and G.A. Slack, *Estimation of the isotope effect on the lattice thermal conductivity of group IV and group III-V semiconductors*. Physical Review B, 2002. **66**(19): p. 195304.
18. Rathnasekara, R., G. Mayberry, and P. Hari, *Thermoelectric, Electrochemical, & Dielectric Properties of Four ZnO Nanostructures*. Materials, 2022. **15**(24): p. 8816.
19. Sulaiman, S., et al., *Review on grain size effects on thermal conductivity in ZnO thermoelectric materials*. RSC advances, 2022. **12**(9): p. 5428-5438.
20. Mulla, R. and M. Rabinal, *Defect-Controlled Copper Iodide: A Promising and Ecofriendly Thermoelectric Material*. Energy Technology, 2018. **6**(6): p. 1178-1185.
21. Liu, A., et al., *Engineering Copper Iodide (CuI) for Multifunctional p-Type Transparent Semiconductors and Conductors*. Advanced Science, 2021. **8**(14): p. 2100546.